# Demonstrating and Reducing Shortcuts in Vision-Language Representation Learning

**Maurits Bleeker**[*]                                    *m.j.r.bleeker@uva.nl*
*University of Amsterdam, Amsterdam, The Netherlands*

**Mariya Hendriksen**[*]                                 *m.hendriksen@uva.nl*
*AIRLab, University of Amsterdam, Amsterdam, The Netherlands*

**Andrew Yates**                                         *a.c.yates@uva.nl*
*University of Amsterdam, Amsterdam, The Netherlands*

**Maarten de Rijke**                                     *m.derijke@uva.nl*
*University of Amsterdam, Amsterdam, The Netherlands*

## Abstract

Vision-language models (VLMs) mainly rely on contrastive training to learn general-purpose representations of images and captions. We focus on the situation when one image is associated with several captions, each caption containing both information shared among all captions and unique information per caption about the scene depicted in the image. In such cases, it is unclear whether contrastive losses are sufficient for learning task-optimal representations that contain all the information provided by the captions or whether the contrastive learning setup encourages the learning of a simple shortcut that minimizes contrastive loss. We introduce *synthetic shortcuts for vision-language*: a training and evaluation framework where we inject synthetic shortcuts into image-text data. We show that contrastive VLMs trained from scratch or fine-tuned with data containing these synthetic shortcuts mainly learn features that represent the shortcut. Hence, contrastive losses are not sufficient to learn task-optimal representations, i.e., representations that contain all task-relevant information shared between the image and associated captions. We examine two methods to reduce shortcut learning in our training and evaluation framework: (i) latent target decoding and (ii) implicit feature modification. We show empirically that both methods improve performance on the evaluation task, but only partially reduce shortcut learning when training and evaluating with our shortcut learning framework. Hence, we show the difficulty and challenge of our shortcut learning framework for contrastive vision-language representation learning.

## 1  Introduction

Recent work on understanding the internal mechanisms of representation learning has brought to attention the problem of shortcut learning (Robinson et al., 2021; Chen et al., 2021; Scimeca et al., 2022). While there are multiple definitions of shortcut learning (e.g., Geirhos et al., 2020; Wiles et al., 2022), in this work we define *shortcuts* as *easy-to-learn discriminatory features that minimize the (contrastive) optimization objective but are not necessarily sufficient for solving the evaluation task*. More specifically, we focus on the problem of shortcut learning in the relatively unexplored context of vision-language (VL) representation learning with multiple matching captions per image.

Contrastive learning (CL) plays a crucial role in VL representation learning. Despite the success of non-contrastive approaches, e.g., (Bardes et al., 2022), the dominant paradigm in VL representation learning revolves around either fully contrastive strategies (Faghri et al., 2018; Li et al., 2019a; Jia et al., 2021;

---

[*]Co-first author.

Radford et al., 2021) or a combination of contrastive methods with additional objectives (Li et al., 2021; Zeng et al., 2022; Li et al., 2022a; Zeng et al., 2022; Li et al., 2023a). It is standard practice in contrastive VL representation learning to sample batches of image-caption pairs and maximize the alignment between the representations of the matching images and captions (Radford et al., 2019; Jia et al., 2021). Given that the typical VL benchmarks, e.g., Flickr30k (Young et al., 2014) and MS-COCO Captions (Lin et al., 2014; Chen et al., 2015), are constructed in such a way that each image is associated with multiple captions, each caption can be seen as a different *view* of the image it describes. Therefore, CL with multiple captions per image can be seen as CL with multiple views, where each caption provides a different view of the scene depicted in the image.

CL with multiple views, where each view represents a different observation of the same datapoint, has proven to be effective for general-purpose representation learning (Hjelm et al., 2019; Chen et al., 2020a; Tian et al., 2020a). The goal of multi-view (contrastive) representation learning methods is to learn representations that remain invariant to a shift of view, which is achieved by maximizing alignment between embeddings of similar views. A core assumption within the multi-view representation learning literature is that task-relevant information is shared across views whereas task-irrelevant information is not shared, given a downstream evaluation task (Zhao et al., 2017; Federici et al., 2020; Tian et al., 2020a; Shwartz-Ziv & LeCun, 2023).

An open challenge in the multi-view representation learning domain concerns *learning representations that contain task-relevant information that is not shared among different views, i.e., that may be unique for some views* (Shwartz-Ziv & LeCun, 2023; Zong et al., 2023). In the case of image-caption datasets where each image is paired with at least one corresponding caption, the captions matching the same image do not necessarily share the same information as each caption is distinct and may describe different aspects of the image (Biten et al., 2022). Figure 1 illustrates the concept of shared vs. caption-specific task-relevant information. The image is accompanied by two captions: 'a couple of boats and a red car' ($\mathbf{x}_{\mathcal{C}_A}$) and 'a couple of boats and a car on a street' ($\mathbf{x}_{\mathcal{C}_B}$). The shared information between the captions includes 'couple of boats' and 'car'. Caption $\mathbf{x}_{\mathcal{C}_A}$ provides unique information by describing the car as 'red'. Caption $\mathbf{x}_{\mathcal{C}_B}$ adds unique contextual details about the location with the phrase 'on a street'. To learn task-optimal representations, it is essential to integrate both the shared and unique information from these captions. Furthermore, given the typical quality of captions of image-caption datasets (Chen et al., 2015), we assume that all information present in the captions is relevant. Hence, each image-caption pair may contain both *shared* task-relevant information, i.e., information shared across all the captions in the tuple, and *unique* task-relevant information, i.e., information not shared with other captions. Therefore, learning task-optimal representations for the image implies learning all task-relevant information that comprises both shared and caption-specific information.

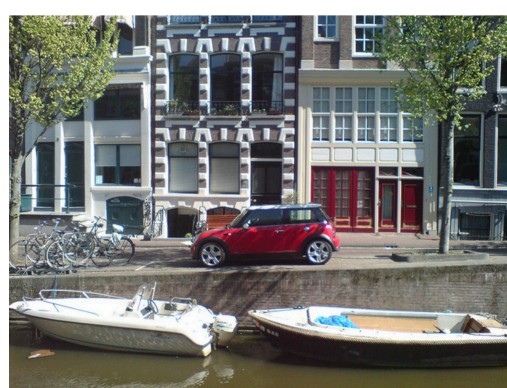

$\mathbf{X}_{\mathcal{C}_A}$ : a couple of boats and a red car

$\mathbf{X}_{\mathcal{C}_B}$ : a couple of boats and car on a street

Figure 1: Shared vs. caption-specific information given an example of one image and two associated captions $\mathbf{x}_{\mathcal{C}_A}$ and $\mathbf{x}_{\mathcal{C}_B}$. The purple color indicates information shared between the image and both captions. The green color indicates task-relevant information specific for $\mathbf{x}_{\mathcal{C}_A}$. The blue color indicates task-relevant information specific for $\mathbf{x}_{\mathcal{C}_B}$.

Another problem of CL approaches is related to *feature suppression*. Shwartz-Ziv & LeCun (2023) argue that although contrastive loss functions lack explicit information-theoretical constraints aimed at suppressing non-shared information among views, the learning algorithm benefits from simplifying representations by suppressing features from the input data that are not relevant for minimizing the contrastive loss. Furthermore, Robinson et al. (2021) demonstrate that contrastive loss functions are susceptible to solutions that suppress features from the input data. In the case of VL, CL with multiple captions per image where at least one caption contains caption-specific information, the image representation can never have a perfect alignment with all matching captions. This is due to the misalignment that happens when encoding unique information for the

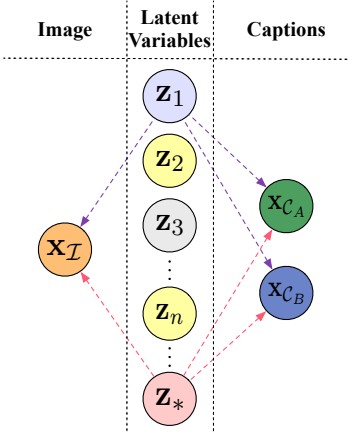
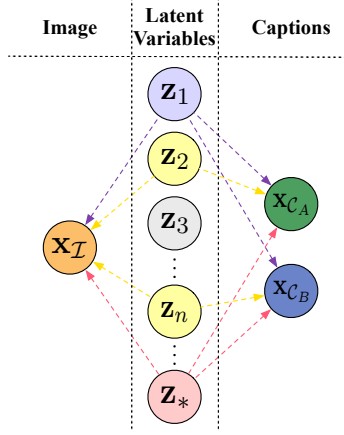

(a) Minimal shared information.

(b) Task-optimal information.

Figure 2: Synthetic shortcuts in the context of minimal shared and task-optimal information for vision-language representation learning with multiple captions per image. The purple color represents features shared among the image and all captions (minimal shared information). The yellow color represents caption-specific features (unique information). The grey color indicates features that are not present in both the image and any of the captions (task-irrelevant information). The red color indicates synthetic shortcuts. We demonstrate that while shortcuts exist in both scenarios, minimal shared information also includes information shared among the image and all associated captions, whereas task-optimal information combines both minimal shared information and caption-specific information.

other captions. Therefore, it is unclear whether contrastive methods can learn task-optimal representations, i.e., representations that contain all information present in the captions associated with the image, or if they learn only the minimal shared information, i.e., information shared between the image and all captions that are sufficient to minimize the contrastive discrimination objective. An illustration of minimal shared information and a task-optimal representation is given in Figure 2.

Motivated by the abovementioned problems, we address the following question:

> *In the context of VL representation learning with multiple captions per image, to what extent does the presence of a shortcut hinder learning task-optimal representations?*

To answer this question, we investigate the problem of shortcut learning for VL representation learning with multiple captions per image. We do this by introducing the *synthetic shortcuts for vision-language* (SVL) framework for adding additional, easily identifiable information to image-caption tuples. The information that we add is represented as identifiers that are applied to both image and caption; these identifiers do not bear any semantic meaning. The identifiers provide additional shared information between the image and captions, which is a subset of the total shared information between the image and the caption. For details and examples of shortcuts, refer to Section 3, where Figure 4 illustrates an example of an image-caption pair with a shortcut added. The synthetic shortcuts framework allows us to investigate how much the encoder model relies on the added shortcut during training and evaluation, and hence how much of the relevant information is still captured if a shortcut solution is available. Overall, our SVL framework allows us to investigate the shortcut learning problem in a controlled way. We focus on image-caption retrieval (ICR) as an evaluation task because contrastive losses directly optimize for the ICR evaluation task, which assesses the quality of the learned representations by computing a similarity score between images and captions (Radford et al., 2021; Yuksekgonul et al., 2023). To investigate the problem, we run experiments on two distinct models: (i) CLIP (Radford et al., 2019), a large-scale model that we fine-tune; and (ii) VSE++ (Faghri et al., 2018), a relatively small model that we train from scratch. We evaluate the models' performance on the Flickr30k (Young et al., 2014) and MS-COCO (Lin et al., 2014; Chen et al., 2015) and benchmarks. The benchmarks are constructed in such a way that each image is associated with five captions and each caption represents a concise summary of the corresponding image.

Therefore, the contributions of this work are two-fold:

I **A framework for investigating the problem of shortcut learning for contrastive vision-language representation learning in a controlled way**: We introduce the *synthetic shortcuts for vision-language* framework. The framework enables the injection of synthetic shortcuts into image-caption tuples in the training dataset. We use the framework to investigate and understand the extent to which contrastive VL models rely on shortcuts when a shortcut solution is available. We run our experiments using CLIP and VSE++, two distinct vision-language models (VLMs). We evaluate the models' performance on the Flickr30k and MS-COCO benchmarks. We evaluate the effectiveness of contrastive VL models by comparing their performance with and without synthetic shortcuts. We demonstrate that both models trained from scratch and fine-tuned, large-scale pre-trained foundation models mainly rely on shortcut features and do not learn task-optimal representations. Consequently, we show that contrastive losses mainly capture the easy-to-learn discriminatory features that are shared among the image and all matching captions, while suppressing other task-relevant information. Hence, we argue that contrastive losses are not sufficient to learn task-optimal representations for VL representation learning.

II **We present two shortcut learning reduction methods on our proposed training and evaluation framework:** We investigate latent target decoding (LTD) and implicit feature modification (IFM) using our SVL training and evaluation framework. While both methods improve performance on the evaluation task, our framework poses challenges that existing shortcut reduction techniques can only partially address, as the performance is not on par with models trained without synthetic shortcuts. These findings underline the importance and complexity of our framework in studying and evaluating shortcut learning within the context of contrastive VL representation learning.

## 2 Background and Analysis

In this section, we present the notation, setup, and assumptions on which we base the work. Additionally, we conduct an analysis of contrastive VL representation learning with multiple captions per image.

### 2.1 Preliminaries

**Notation.** We closely follow the notation from (Bleeker et al., 2023). See Table 3 for an overview. Let $\mathcal{D}$ be a dataset of $N$ image-caption tuples: $\mathcal{D} = \left\{ \left( \mathbf{x}_\mathcal{I}^i, \{\mathbf{x}_{\mathcal{C}_j}^i\}_{j=1}^k \right) \right\}_{i=1}^N$. Each tuple $i \in N$ contains one image $\mathbf{x}_\mathcal{I}^i$ and $k$ captions $\mathbf{x}_{\mathcal{C}_j}^i$, where $1 \le j \le k$. All captions in tuple $i \in N$ are considered as matching captions w.r.t. image $\mathbf{x}_\mathcal{I}$ in the tuple $i$. The latent representation of an image-caption pair from a tuple $i$ is denoted as $\mathbf{z}_\mathcal{I}^i$ and $\mathbf{z}_{\mathcal{C}_j}^i$ respectively. During training, we sample image-caption pairs from the dataset $\mathcal{D}$ and optimize for the evaluation task $T$. We include all captions in the dataset once per training epoch, hence, each image is sampled $k$ times.

Given an image $\mathbf{x}_\mathcal{I}$, a set of $k$ associated captions $K = \{\mathbf{x}_{\mathcal{C}_j}\}_{j=1}^k$, and one caption randomly sampled from the set $\mathbf{x}_\mathcal{C} \in K$, we define the following representations: (i) $\mathbf{z}_{\mathcal{C} \to \mathcal{I}}^{SUF}$ as *sufficient* representation of the caption $\mathbf{x}_\mathcal{C}$ that describes the image $\mathbf{x}_\mathcal{I}$; (ii) $\mathbf{z}_{\mathcal{I} \to \mathcal{C}}^{SUF}$ as representation of the image $\mathbf{x}_\mathcal{I}$ *sufficient for the caption* $\mathbf{x}_\mathcal{C}$; (iii) $\mathbf{z}_{\mathcal{I} \to \mathcal{C}}^{MIN}$ as representation of the image $\mathbf{x}_\mathcal{I}$ that is *minimally sufficient for the caption* $\mathbf{x}_\mathcal{C}$; and (iv) $\mathbf{z}_{\mathcal{I} \to K}^{OPT}$ as representation of the image $\mathbf{x}_\mathcal{I}$ that is *optimal for the set of captions* $K$ given the task $T$.

In addition, we write $S_{SynSC}$ for a synthetic shortcut, $S$ for the original shared information, i.e., information that does not contain synthetic shortcuts, $S^+$ for the shared information that includes a synthetic shortcut, and $R^+$ for task-relevant information that contains a synthetic shortcut.

In the context of task relevance, we define $R$ and $\neg R$ as task-relevant and task-irrelevant information, respectively, and $C$ as task-relevant information specific for caption $\mathbf{x}_\mathcal{C}$.

**Setup.** We work with a dual-encoder setup, with an image encoder and a caption encoder that do not share parameters. The *image encoder* $f_\theta(\cdot)$ takes an image $\mathbf{x}_\mathcal{I}$ as input and returns its latent representation: $\mathbf{z}_\mathcal{I} := f_\theta(\mathbf{x}_\mathcal{I})$. Similarly, the *caption encoder* $g_\phi(\cdot)$ takes a caption $\mathbf{x}_\mathcal{C}$ as input, and encodes the caption into a latent representation: $\mathbf{z}_\mathcal{C} := g_\phi(\mathbf{z}_\mathcal{C})$. Both $\mathbf{z}_\mathcal{C}$ and $\mathbf{z}_\mathcal{I}$ are unit vectors projected into $d$-dimensional multi-modal space: $\mathbf{z}_\mathcal{C} \in \mathbb{R}^d$, $\mathbf{z}_\mathcal{I} \in \mathbb{R}^d$. For an overview of notation, we refer to Appendix A, Table 3.

**Assumptions.** Given an image-caption tuple, we assume that each caption in the tuple is distinct from the other captions in the tuple. We also assume that each caption in the tuple contains two types of task-relevant information: (i) shared information, i.e., information shared with other captions in the same tuple, and (ii) caption-specific information, i.e., information that is not shared with the other captions. For simplicity, we base our subsequent analysis on tuples where one image $\mathbf{x}_\mathcal{I}$ is associated with two captions $\mathbf{x}_{\mathcal{C}_A}$ and $\mathbf{x}_{\mathcal{C}_B}$: $\left(\mathbf{x}_\mathcal{I}, \{\mathbf{x}_{\mathcal{C}_A}, \mathbf{x}_{\mathcal{C}_B}\}\right)$. However, the analysis described in this section can be extended to a case with more than two captions. We treat images and captions as views and define $\mathbf{x}_\mathcal{I}$, $\mathbf{x}_{\mathcal{C}_A}$, and $\mathbf{x}_{\mathcal{C}_B}$ to be random variables of an image and two matching captions, with the joint distribution $p(\mathbf{x}_\mathcal{I}, \mathbf{x}_{\mathcal{C}_A}, \mathbf{x}_{\mathcal{C}_B})$. For more details on assumptions and problem definition, we refer to Appendix B.

## 2.2 Analysis of Contrastive Vision-Language Representation Learning for Multiple Captions per Image

**InfoMax.** We start our analysis of contrastive VL representation learning by introducing the InfoMax optimization objective, a typical loss for VL representation learning. The goal of an InfoMax optimization objective, e.g., InfoNCE (van den Oord et al., 2018), is to maximize the mutual information (MI) between the latent representations of two views of the same data (Tschannen et al., 2020). Therefore, the optimization objective is equivalent to: $\max_{f_\theta, g_\phi} I(\mathbf{z}_\mathcal{I}; \mathbf{z}_\mathcal{C})$ where $\mathbf{z}_\mathcal{I} = f_\theta(\mathbf{x}_\mathcal{I})$ and $\mathbf{z}_\mathcal{C} := g_\phi(\mathbf{x}_\mathcal{C})$.

**Minimally Sufficient Image Representation.** During training, batches of image-caption pairs are sampled. The optimization involves maximizing the MI between the image representation $\mathbf{z}_\mathcal{I}$ and the matching caption representation $\mathbf{z}_\mathcal{C}$. Wang et al. (2022) argue that, since all supervision information for one view (i.e., the image) comes from the other view (i.e., the caption), the representations learned contrastively are approximately minimally sufficient. Following (Tian et al., 2020b; Wang et al., 2022), we extend the definition of sufficient representation to VL context and define sufficient caption representations, sufficient image representations, and minimally sufficient image representation.

**Definition 2.1** (Sufficient caption representation). *Given an image $\mathbf{x}_\mathcal{I}$, and a set of matching captions $\mathcal{C} = \{\mathbf{x}_{\mathcal{C}_A}, \mathbf{x}_{\mathcal{C}_B}\}$, the representation $\mathbf{z}_{\mathcal{C} \to \mathcal{I}}^{SUF}$ of caption $\mathbf{x}_\mathcal{C} \in \mathcal{C}$ is sufficient for image $\mathbf{x}_\mathcal{I}$ if, and only if, $I(\mathbf{z}_{\mathcal{C} \to \mathcal{I}}^{SUF}; \mathbf{x}_\mathcal{I}) = I(\mathbf{x}_\mathcal{C}; \mathbf{x}_\mathcal{I})$.*

The sufficient caption representation $\mathbf{z}_{\mathcal{C} \to \mathcal{I}}^{SUF}$ contains all the information about image $\mathbf{x}_\mathcal{I}$ in caption $\mathbf{x}_\mathcal{C}$.

**Definition 2.2** (Sufficient image representation). *Given an image $\mathbf{x}_\mathcal{I}$, and a set of matching captions $\mathcal{C} = \{\mathbf{x}_{\mathcal{C}_A}, \mathbf{x}_{\mathcal{C}_B}\}$, the representation $\mathbf{z}_{\mathcal{I} \to \mathcal{C}}^{SUF}$ of image $\mathbf{x}_\mathcal{I}$ is sufficient for caption $\mathbf{x}_\mathcal{C} \in \mathcal{C}$ if, and only if, $I(\mathbf{z}_{\mathcal{I} \to \mathcal{C}}^{SUF}; \mathbf{x}_\mathcal{C}) = I(\mathbf{x}_\mathcal{I}; \mathbf{x}_\mathcal{C})$.*

Similarly, the sufficient image representation $\mathbf{z}_{\mathcal{I} \to \mathcal{C}}^{SUF}$ contains all the shared information between an image $\mathbf{x}_\mathcal{I}$ and a caption $\mathbf{x}_\mathcal{C}$. Note that a sufficient image representation can be sufficient w.r.t. multiple captions.

**Definition 2.3** (Minimally sufficient image representation). *Given an image $\mathbf{x}_\mathcal{I}$, and a set of matching captions $\mathcal{C} = \{\mathbf{x}_{\mathcal{C}_A}, \mathbf{x}_{\mathcal{C}_B}\}$, the sufficient image representation $\mathbf{z}_{\mathcal{I} \to \mathcal{C}}^{MIN}$ of image $\mathbf{x}_\mathcal{I}$ is minimally sufficient for caption $\mathbf{x}_\mathcal{C} \in \mathcal{C}$ if, and only if, $I(\mathbf{z}_{\mathcal{I} \to \mathcal{C}}^{MIN}; \mathbf{x}_\mathcal{I}) \leq I(\mathbf{z}_{\mathcal{I} \to \mathcal{C}}^{SUF}; \mathbf{x}_\mathcal{I})$, for all $\mathbf{z}_{\mathcal{I} \to \mathcal{C}}^{SUF}$ that are sufficient.*

Intuitively, $\mathbf{z}_{\mathcal{I} \to \mathcal{C}}^{MIN}$ comprises the smallest amount of information about $\mathbf{x}_\mathcal{I}$ (while still being sufficient) and, therefore, only contains the information that is shared with caption $\mathbf{x}_\mathcal{C}$, i.e., the non-shared information is suppressed.

**Task-Optimal Image Representation.** The definition of task-optimal image representation is based on the notion of task-relevant information. In the context of VL representation learning with multiple captions per image, we define task-relevant information as all information described by the matching captions. That

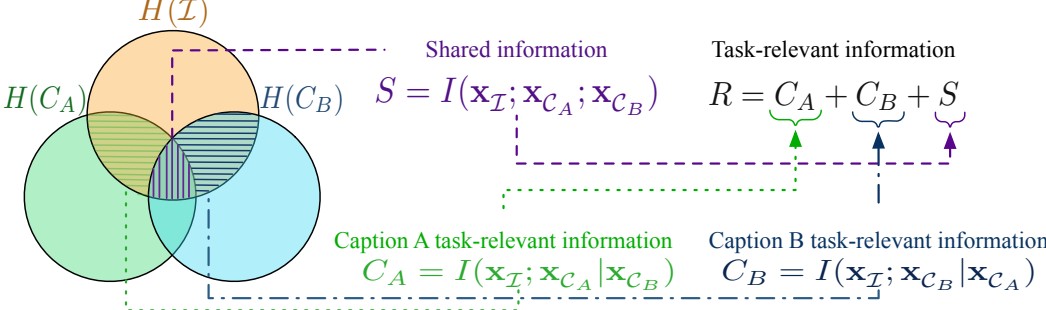

Figure 3: We define $H(\mathbf{x}_{\mathcal{I}})$ as image information, $H(\mathbf{x}_{\mathcal{C}_A})$ and $H(\mathbf{x}_{\mathcal{C}_B})$ as caption information; both captions only describe the information depicted in the image and contain shared and caption-specific information. We further define $C_A = I(\mathbf{x}_{\mathcal{I}}; \mathbf{x}_{\mathcal{C}_A} \mid \mathbf{x}_{\mathcal{C}_B})$ and $C_B = I(\mathbf{x}_{\mathcal{I}}; \mathbf{x}_{\mathcal{C}_B} \mid \mathbf{x}_{\mathcal{C}_A})$ as caption-specific information; $S = I(\mathbf{x}_{\mathcal{I}}; \mathbf{x}_{\mathcal{C}_A}; \mathbf{x}_{\mathcal{C}_B})$ as shared information; $\neg R = H(\mathbf{x}_{\mathcal{I}} \mid \mathbf{x}_{\mathcal{C}_A}, \mathbf{x}_{\mathcal{C}_B})$ as task-irrelevant information; $R = C_A + C_B + S$ as task-relevant information.

includes both caption-specific and shared information. Consequently, task-optimal image representation is image representation that is sufficient w.r.t. all matching captions.

Formally, following assumptions from Appendix B.2, we define task-relevant information $R$ as all the information described by the matching captions. The task-relevant information can be expressed as follows:

$$
\underbrace{R}_{\substack{\text{Task-relevant} \\ \text{information}}} = \underbrace{H(\mathbf{x}_{\mathcal{I}})}_{\substack{\text{Image} \\ \text{information}}} - \underbrace{H(\mathbf{x}_{\mathcal{I}} \mid \mathbf{x}_{\mathcal{C}_A}, \mathbf{x}_{\mathcal{C}_B})}_{\substack{\text{Task-irrelevant} \\ \text{information}}}
$$
$$
= \underbrace{I(\mathbf{x}_{\mathcal{I}}; \mathbf{x}_{\mathcal{C}_A} \mid \mathbf{x}_{\mathcal{C}_B})}_{\substack{C_A\text{-specific} \\ \text{task-relevant information}}} + \underbrace{I(\mathbf{x}_{\mathcal{I}}; \mathbf{x}_{\mathcal{C}_B} \mid \mathbf{x}_{\mathcal{C}_A})}_{\substack{C_B\text{-specific} \\ \text{task-relevant information}}} + \underbrace{I(\mathbf{x}_{\mathcal{I}}; \mathbf{x}_{\mathcal{C}_A}; \mathbf{x}_{\mathcal{C}_B})}_{\substack{\text{Shared} \\ \text{information}}}. \tag{1}
$$

Similarly, task-irrelevant information $\neg R$ is the image information not described by the captions. Figure 3 illustrates both definitions.

The multi-view assumption states that task-relevant information for downstream tasks comes from the information shared between views (Shwartz-Ziv & LeCun, 2023). However, in the case of VL representation learning with multiple captions per image, task-relevant information $R$ includes both shared information $S$, and caption-specific information $C_A$ and $C_B$ (Eq. 1).

**Definition 2.4** (Task-optimal image representation). *Given an image $\mathbf{x}_{\mathcal{I}}$, and a set of matching captions $\mathcal{C} = \{\mathbf{x}_{\mathcal{C}_A}, \mathbf{x}_{\mathcal{C}_B}\}$, the representation $\mathbf{z}_{\mathcal{I} \rightarrow \mathcal{C}}^{OPT}$ is task-optimal image representation for all matching captions if, and only if, $I(\mathbf{z}_{\mathcal{I} \rightarrow \mathcal{C}}^{OPT}; \mathbf{x}_{\mathcal{C}}) = I(\mathbf{x}_{\mathcal{I}}; \mathbf{x}_{\mathcal{C}})$, for all $\mathbf{x}_{\mathcal{C}} \in \mathcal{C}$.*

In other words, task-optimal image representations contain all the information that the image shares with the matching captions. Hence, a task-optimal image representation is sufficient w.r.t. all matching captions. The information contained in the task-optimal image representation includes both shared and caption-specific information. Therefore, a task-optimal image representation can never be a minimally sufficient image representation w.r.t. to a specific caption.

**Theorem 1** (Suboptimality of contrastive learning with multiple captions per image). *Given an image $\mathbf{x}_{\mathcal{I}}$, a set of matching captions $\mathcal{C} = \{\mathbf{x}_{\mathcal{C}_A}, \mathbf{x}_{\mathcal{C}_B}\}$, and a contrastive learning loss function $\mathcal{L}_{InfoNCE}$ that optimizes for task $T$, image representations learned during contrastive learning will be minimally sufficient and will never be task-optimal image representations.*

The proof is provided in Appendix C. Rephrasing Theorem 1, given an image and two captions that form two image-caption pairs, $(\mathbf{x}_{\mathcal{I}}, \mathbf{x}_{\mathcal{C}_A})$ and $(\mathbf{x}_{\mathcal{I}}, \mathbf{x}_{\mathcal{C}_B})$, and assuming that contrastive loss optimizes the image encoder to be minimally sufficient w.r.t. to caption $\mathbf{x}_{\mathcal{C}_A}$ during a training step, all task-relevant information $C_B$ specific to caption $\mathbf{x}_{\mathcal{C}_B}$ will be suppressed in $\mathbf{z}_{\mathcal{I}}$. Hence, the resulting image representation will not be optimal for the task $T$.

Theorem 1 indicates a discrepancy between minimally sufficient representations learned during contrastive training with the InfoNCE loss and the task-optimal image representations in the context of learning VL representations with multiple captions per image. Although the InfoMax loss does not have an explicit constraint to compress information, prior work indicates that feature suppression is happening (Shwartz-Ziv & LeCun, 2023; Robinson et al., 2021). Hence, we question if contrastive loss can be used to learn task-optimal image representations in the context of multiple captions per image.

Furthermore, Theorem 1 implies that in the context of contrastive VL representation learning with multiple captions per image, the minimally sufficient representation, which discards non-shared information, is not the same as the task-optimal representation that comprises both caption-specific and shared information. This suggests that the features learned during contrastive learning might be shortcuts, i.e., easy-to-detect discriminatory features that minimize the contrastive optimization objective but are not necessarily sufficient for solving the evaluation task. To examine this problem, we introduce a synthetic shortcuts framework that allows us to investigate the problem of suboptimality of contrastive learning with multiple captions per image in a controlled way.

## 3    Synthetic Shortcuts to Control Shared Information

In Section 2 we show the suboptimality of the contrastive InfoNCE loss with multiple captions per image. In the case of real-world VL datasets with multiple captions per image, there are no annotations that indicate the information shared between the image and captions and the information specific to each caption. Hence, we cannot directly measure how much of the shared and unique information is captured by the representations.

**Synthetic Shortcuts.** In this section, we introduce the *synthetic shortcuts for vision-language (SVL)* training and evaluation framework. We denote the *synthetic shortcuts for image-caption data* as $S_{SynSC}$. The purpose of the framework is to introduce additional and easily identifiable information shared between an image and the matching captions that lacks any semantic meaning. The shortcuts we use in this work are represented as numbers that we add to images and captions. For images, we add the shortcut number by adding MNIST images as an overlay to the original images. For captions, we append the numbers of the shortcut as extra tokens at the end of the caption.

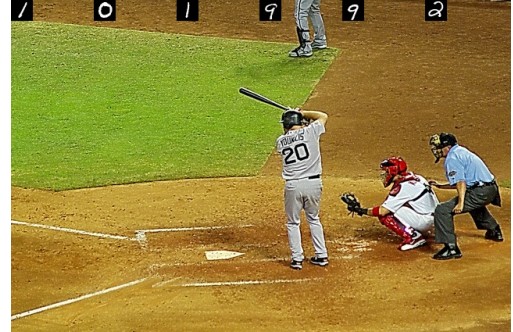

A player up to bat in a baseball game. 1 0 1 9 9 2

Figure 4: An image-caption pair from the MS-COCO dataset with a shortcut added to both the image and the caption.

Figure 4 illustrates an example of an image-caption pair with an added shortcut. The example contains an image with the caption: 'A player up to bat in a baseball game. 1 0 1 9 9 2.' Here, '1 0 1 9 9 2' is a shortcut added to both the image and the caption. For the image modality, we add the shortcut by overlaying MNIST images at the top of the original image. For the text modality, we append the shortcut as additional tokens at the end of the caption. This identifier provides an additional link between the image and the caption without carrying any semantic meaning related to their content. Additional examples are shown in Figure 6.

If contrastive losses learn task-optimal representations, then the presence of synthetic shortcuts should not negatively impact the evaluation performance, since synthetic shortcuts represent additional information and the remaining task-relevant information is intact. By incorporating synthetic shortcuts into the image-caption dataset, the shared information would include the information that was originally shared and the synthetic shortcut: $S^+ = S + S_{SynSC}$. Hence, the task-relevant information would comprise caption-specific information that was originally shared and a synthetic shortcut: $R^+ = C_A + C_B + S + S_{SynSC}$. If injecting a synthetic shortcut influences the performance negatively, we can conclude that by learning to represent a synthetic shortcut the model suppresses other task-relevant information in favor of the shortcut, hence the representation

is not task-optimal. The setup is inspired by the "datasets with explicit and controllable competing features," introduced by Chen et al. (2021), but we adapt this setup to the VL scenario.

For experiments, we use the Flickr30k and MS-COCO image-caption datasets, that consist of image-caption tuples, each image is associated with five captions. During training, we sample a batch of image-caption pairs $\mathcal{B} = \{(\mathbf{x}_{\mathcal{I}}^i, \mathbf{x}_{\mathcal{C}_j}^i), \dots\}_{i=1}^{|\mathcal{B}|}$, from dataset $\mathcal{D}$, and apply shortcut sampling. We inject the shortcuts in a manner that preserves the original information of the images and captions. Furthermore, we append the shortcut after applying data augmentations to ensure that the shortcut is present in both the images and captions (i.e., the shortcut is not augmented away). We refer to Figure 6 for some examples. The training, evaluation, and implementation details of the shortcut sampling are provided in Appendix D.4.

We define the following experimental setups:

I *No shortcuts*: As a baseline, we fine-tune a pre-trained CLIP (Radford et al., 2021) and train VSE++ (Faghri et al., 2018) from scratch on Flickr30k and MS-COCO, without using any shortcuts. The experimental setup for training both models is provided in Appendix D.2 and D.3. The goal of this setup is to show the retrieval evaluation performance without adding any shortcuts for both a large-scale pre-trained foundation model and a small-scale model trained from scratch.

II *Unique shortcuts*: We add a unique shortcut to each image-caption tuple $i \in \mathcal{D}$ in the dataset. In this setup, each image caption pair can be uniquely matched during training by only detecting the shortcut. For each tuple $i \in \mathcal{D}$, we use the number $i$ as the number of the shortcut we inject to the image and captions in the tuple. If the contrastive loss learns task-optimal representations, the downstream evaluation performance should not decrease when training with unique shortcuts.

III *Unique shortcuts on only one modality*: To show that the shortcuts do not interfere with the original task-relevant information ($S, C_A$, and $C_B$) of the images and captions, we create a dataset with only shortcuts on either the image or caption modality. Therefore, the shortcut cannot be used by the encoders to match an image-caption pair. Hence, we expect the encoders to ignore the shortcuts and extract the features from the original data similar to the features learned by the baseline models in experimental setup I.

IV *N bits of shortcuts*: In this setup, for each image-caption pair in the training batch $\mathcal{B}$, we randomly sample a shortcut number from the range $[0, 2^n]$, where $n$ is the number of bits. The higher the value of $n$, the more image-caption pairs in the training batch will have by expectation a unique shortcut, and, the less the model has to rely on $S$ and the remaining task-relevant information to solve the contrastive objective. The goal of this setup is to show that, the more unique (shortcut) information is present per sample in the batch, the less contrastive models rely on the remaining task-relevant information.

It should be noted that the shortcuts we add are independent of the image-caption pairs. However, the goal of the SVL framework is to measure the effect of the presence of additional easy-to-detect shared information on the learned representations.

**Evaluation Method.** To show the effect of the injected shortcuts on retrieval evaluation performance, we evaluate both with and without adding the shortcuts during evaluation. When training with unique shortcuts, we add a unique shortcut to each tuple in the test set as well. When training with shortcuts on either one of the two modalities, we only evaluate without shortcuts to show that training with shortcuts on one modality does not influence performance. When training with $n$ bits of shortcuts, we add the shortcut $\mod (i, n)$ (modulo) to each tuple $i$ in the evaluation set, to make sure we use the same number of shortcuts during evaluation as during training. To facilitate the reproducibility and support further research, we provide the code with our paper.[1]

---

[1] https://github.com/MauritsBleeker/svl-framework

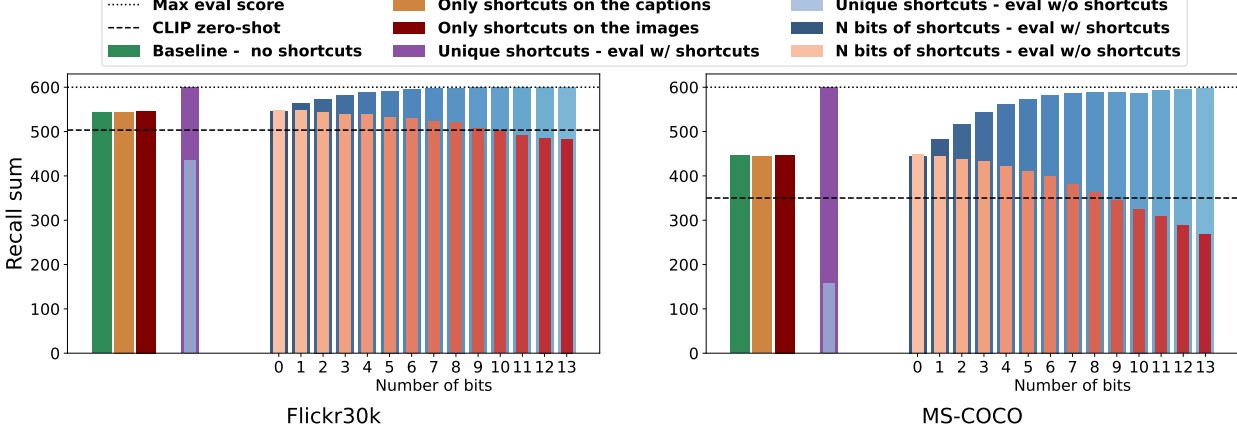

(a) Evaluation results for the CLIP model when using different shortcut sampling setups.

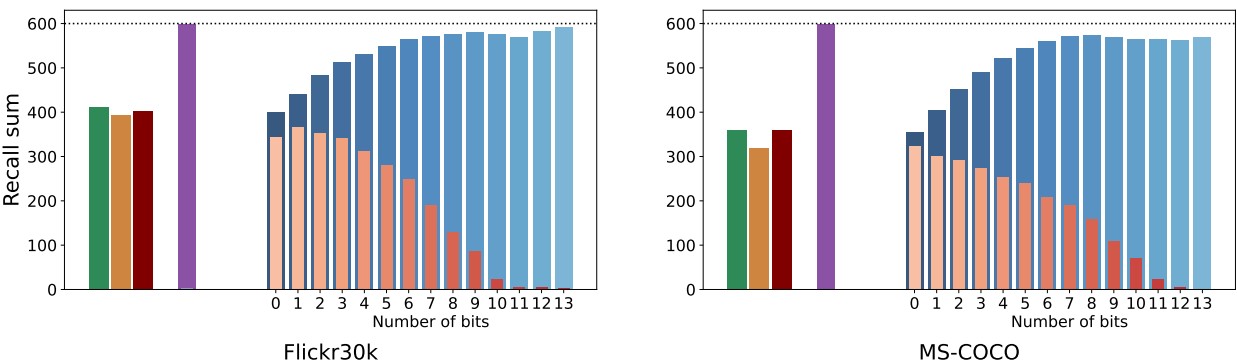

(b) Evaluation results for the VSE++ model when using different shortcut sampling setups.

Figure 5: Effect of synthetic shortcuts on CLIP and VSE++ performance on ICR task. The dotted line represents the maximum achievable recall sum, while the dashed line for CLIP indicates its zero-shot evaluation performance (Best viewed in color.)

## 4 Synthetic Shortcuts and their Impact on Learned Representations and Evaluation Performance

### 4.1 Findings

First, we train and evaluate both a CLIP and VSE++ without shortcuts on the Flickr30k and MS-COCO dataset for the image-caption retrieval task as a baseline. We use the recall sum (i.e., the sum of $R@1$, $R@5$, and $R@10$ for both image-to-text (i2t) and text-to-image (t2i) retrieval) as evaluation metric (see Appendix B.1 for the evaluation task description). We visualize the results in Figure 5. The dotted line (in Figure 5a and 5b) indicates the maximum evaluation score (i.e., 600). For CLIP, we also provide the zero-shot performance of the model, indicated by the dashed line in Figure 5a. When referring to specific results in Figure 5, we use the color of the corresponding bar and legend key in brackets in the text.

Based on Figure 5, we draw the following conclusions:

I When training CLIP and VSE++ with only shortcuts on either the caption modality (in Figure 5, the corresponding bar/legend box is colored �) or on the image modality ( ■ , in Figure 5), we do not observe a drop in evaluation scores for CLIP compared to the baseline model ( ■ , in Figure 5a). For VSE++ we only observe a slight drop in evaluation score when training with shortcuts on the caption modality (again ▬ , mainly for MS-COCO, in Figure 5b). Therefore, we conclude that the

synthetic shortcuts do not interfere with the original shared information $S$ or other task-relevant information.

II When training the models with *unique shortcuts*, we observe for both CLIP and VSE++ that when evaluating with shortcuts ( ▬ , in Figure 5), the models obtain a perfect evaluation score. When evaluating without shortcuts ( ▬ , in Figure 5) the evaluation score for VSE++ drops to zero and for CLIP below the zero-shot performance. We conclude that with unique shortcuts: (i) both CLIP and VSE++ fully rely on the shortcuts to solve the evaluation task, (ii) VSE++ has not learned any other shared or task-relevant information other than the shortcuts (since it is trained from scratch, only detecting the shortcuts is sufficient to minimize the contrastive loss), and (iii) fine-tuned CLIP has suppressed original features from the zero-shot model in favor of the shortcuts.

III When training the models with *N bits of shortcuts*, we observe for both CLIP and VSE++ that the larger the number of bits we use during training and when evaluating without shortcuts ( ▬ , in Figure 5), the bigger the drop in evaluation performance. When we evaluate with shortcuts ( ▬ , in Figure 5), the evaluation performance improves as we use more bits compared to the baseline without shortcuts ▬ , in Figure 5). For VSE++, evaluating without shortcuts ( ▬ , in Figure 5b) results in a drop to zero when having a large number of bits. For CLIP, the evaluation performance drops below the zero-shot performance. If we train with 0 bits of shortcuts (i.e., the shortcut is a constant) we do not observe any drop or increase in evaluation scores for CLIP.

## 4.2 Upshot

Given the findings based on Figure 5 we conclude that a contrastive loss (i.e., InfoNCE) mainly learns the easy-to-detect minimal shared features among image-caption pairs that are sufficient to minimize the contrastive objective while suppressing the remaining shared and/or task-relevant information. If contrastive losses are sufficient to learn task-optimal representations for image-caption matching, these shortcuts should not adversely impact the evaluation performance. Moreover, if the contrastive loss would only learn features that are shared among the image and all captions (i.e, $S$), we should not observe a drop in performance to 0 for the VSE++ model when training with unique shortcuts, since there is still a lot of task-relevant information present in $S$. Especially in a training setup where a model is trained from scratch or fine-tuned on small datasets, the easy-to-detect features are likely not equivalent to all task-relevant information in the images and captions. Hence, we conclude that contrastive loss itself is not sufficient to learn task-optimal representations of the images (and sufficient representations of captions) and that it only learns the minimal easy-to-detect features that are needed to minimize the contrastive objective.

## 5 Reducing Shortcut Learning

In the earlier section, we have demonstrated that contrastive loss mainly relies on the minimal, easy-to-detect features shared among image-caption pairs while suppressing remaining task-relevant information. In this section, we describe two methods that help to reduce shortcut learning for contrastive learning on our SVL framework: Latent target decoding (Bleeker et al., 2023) and implicit feature modification (Robinson et al., 2021).

## 5.1 Latent Target Decoding

Latent target decoding (LTD) (Bleeker et al., 2023) is a method to reduce predictive feature suppression (i.e., shortcut learning) for resource-constrained contrastive image-caption matching. The contrastive objective (i.e., InfoNCE) is combined with an additional reconstruction loss, which reconstructs the input caption from the latent representation of the caption $\mathbf{z}^i_{\mathcal{C}_j}$. We refer to Appendix E.2 for the mathematical definition of LTD. Instead of reconstructing the tokens of the input caption in an auto-regressive manner (i.e., auto-encoding), the caption is reconstructed non-auto-regressively, by mapping the caption representation into the latent space of a Sentence-BERT (Reimers & Gurevych, 2019; Song et al., 2020) and minimizing the distance (i.e., reconstructing) between the reconstruction and the Sentence-BERT representation of the caption $\mathbf{x}^i_{\mathcal{C}_j}$. The assumption is that the *target* generated by the Sentence-BERT model contains all task-relevant information

in the caption. Hence, by correctly mapping the latent caption representation $\mathbf{z}_{\mathcal{C}_j}^i$ into the latent space of Sentence-BERT, the caption encoder cannot suppress any task-relevant information or rely on shortcut solutions. LTD is implemented both as a dual-loss objective (i.e., the contrastive loss and LTD are added up) and as an optimization constraint while minimizing the InfoNCE loss, by implementing the loss as a Lagrange multiplier. For the mathematical definition of LTD, we refer to Appendix E.2.

**Experimental Setup.** We use the LTD implementation and set-up similar to Bleeker et al. (2023). We train both CLIP and VSE++ with LTD, implemented as either dual loss or an optimization constraint. When implementing LTD as a constraint, we try $\eta \in \{0.01, 0.05, 0.1, 0.15, 0.2, 0.25, 0.3\}$ as bound values. Similar to Bleeker et al. (2023), when implementing LTD as a dual loss, we use $\beta = 1$ as balancing parameters. We train both with and without unique shortcuts. We do this to show (i) what the performance improvement is compared to using only InfoNCE, and (ii) to what degree LTD prevents full collapse to shortcut features. For each model and dataset, we take the training setup that results in the highest performance on the validation set.

## 5.2 Implicit Feature Modification

Implicit feature modification (IFM) (Robinson et al., 2021) is a method, originally introduced in the context of representation learning for images, that applies perturbations to logits used for guiding contrastive models. IFM perpetuates features that the encoders use during a training step to discriminate between positive and negative samples. By doing so, IFM alters the features that are currently used to solve the discrimination task, to avoid the InfoNCE loss to learn shortcut solutions. How much of the features are removed, is defined by a perturbation budget $\epsilon$. IFM is implemented as a dual loss in combination with the InfoNCE loss. For the mathematical definition of IFM, we refer to Appendix E.3.

**Experimental Setup.** We apply a similar experimental set-up for IFM as for LTD. We apply IFM both to CLIP and to VSE++, both with and without unique shortcuts. Similar to (Robinson et al., 2021), we try different perturbation budgets $\epsilon$, we try $\epsilon \in \{0.05, 0.1, 0.2, 0.5, 1\}$. In line with the LTD setup, we take the training setup that results in the highest performance on the validation set.

## 5.3 Method Comparison

Both LTD and IFM aim to mitigate shortcut learning through different approaches. LTD aims to learn all task-relevant information by reconstructing the input captions. In contrast, IFM perturbs the discriminative features in the latent space of the encoder and does not rely on a reconstruction objective. Overall, both methods represent distinct strategies for improving the robustness and generalization capabilities of VL representation learning.

In the following section, we present experimental results with LTD and IFM, providing insight into their effectiveness in mitigating shortcut learning.

# 6 Experimental Results

## 6.1 Does Latent Target Decoding Reduce Shortcut Learning?

In Table 1 we summarize the effect of LTD on reducing shortcut learning.

For CLIP, for both the Flickr30k and MS-COCO dataset, we do not observe an increase in recall scores when fine-tuning with $\mathcal{L}_{\text{InfoNCE+LTD}}$ compared to models that are only fine-tuned with $\mathcal{L}_{\text{InfoNCE}}$. LTD has originally been proposed for resource-constrained VL models. We argue that the additional features that LTD can extract are either already present in the pre-trained CLIP model, or not relevant for the evaluation task. However, when fine-tuning with $\mathcal{L}_{\text{InfoNCE+LTD}}$ and in the presence of shortcuts in the training data, degradation in recall scores is significantly lower than when fine-tuned only with the $\mathcal{L}_{\text{InfoNCE}}$. This shows that LTD can reduce the suppression of features in favor of the shortcut features when fine-tuning large-scale VL models.

Table 1: Mean and variance (over three training runs) recall@$k$ evaluation scores for the Flickr30k and MS-COCO datasets for image-to-text and text-to-image retrieval. We train with two loss functions: $\mathcal{L}_{\text{InfoNCE}}$ and $\mathcal{L}_{\text{InfoNCE+LTD}}$. We train either with (✓) or without (✗) shortcuts. For the model trained with $\mathcal{L}_{\text{InfoNCE+LTD}}$, we provide the hyper-parameters of the best-performing model. $\eta$ indicates that the best-performing model uses LTD implemented as an optimization constraint with bound $\eta$. $\beta$ indicates that the best-performing model uses LTD implemented as a dual-loss with $\beta = 1$.

| Loss | $S_{SynSC}$ | i2t | | | t2i | | | rsum |
|---|---|---|---|---|---|---|---|---|
| | | R@1 | R@5 | R@10 | R@1 | R@5 | R@10 | |
| | | | | Flickr30k | | | | |
| | | | | CLIP | | | | |
| $\mathcal{L}_{\text{InfoNCE}}$ | ✗ | $86.9_{\pm 0.1}$ | $\mathbf{97.4}_{\pm 0.1}$ | $\mathbf{99.0}_{\pm 0.0}$ | $72.4_{\pm 0.1}$ | $\mathbf{92.1}_{\pm 0.0}$ | $\mathbf{95.8}_{\pm 0.0}$ | $543.5_{\pm 1.1}$ |
| $\mathcal{L}_{\text{InfoNCE+LTD}}, \beta = 1$ | ✗ | $86.5_{\pm 0.6}-$ | $97.1_{\pm 0.0}\downarrow$ | $98.5_{\pm 0.0}\downarrow$ | $72.4_{\pm 0.0}-$ | $92.3_{\pm 0.0}\downarrow$ | $95.9_{\pm 0.0}\downarrow$ | $542.8_{\pm 0.8}-$ |
| $\mathcal{L}_{\text{InfoNCE}}$ | ✓ | $57.2_{\pm 8.3}$ | $84.0_{\pm 4.8}$ | $91.0_{\pm 1.9}$ | $44.9_{\pm 4.5}$ | $74.9_{\pm 6.0}$ | $84.2_{\pm 2.5}$ | $436.2_{\pm 145.0}$ |
| $\mathcal{L}_{\text{InfoNCE+LTD}}, \beta = 1$ | ✓ | $\mathbf{64.0}_{\pm 1.3}\uparrow$ | $\mathbf{87.8}_{\pm 0.9}\uparrow$ | $\mathbf{93.2}_{\pm 0.8}\uparrow$ | $\mathbf{50.7}_{\pm 0.6}\uparrow$ | $\mathbf{79.8}_{\pm 0.7}\uparrow$ | $\mathbf{88.1}_{\pm 0.5}\uparrow$ | $\mathbf{463.6}_{\pm 17.3}\uparrow$ |
| | | | | VSE++ | | | | |
| $\mathcal{L}_{\text{InfoNCE}}$ | ✗ | $52.6_{\pm 1.1}$ | $79.8_{\pm 0.1}$ | $87.8_{\pm 0.1}$ | $39.5_{\pm 0.3}$ | $69.8_{\pm 0.0}$ | $79.4_{\pm 0.1}$ | $409.0_{\pm 4.0}$ |
| $\mathcal{L}_{\text{InfoNCE+LTD}}, \eta = 0.2$ | ✗ | $\mathbf{54.1}_{\pm 0.1}\uparrow$ | $\mathbf{81.1}_{\pm 0.8}\uparrow$ | $\mathbf{88.6}_{\pm 0.1}\uparrow$ | $\mathbf{42.5}_{\pm 0.0}\uparrow$ | $\mathbf{71.9}_{\pm 0.1}\uparrow$ | $\mathbf{81.3}_{\pm 0.0}\uparrow$ | $\mathbf{419.6}_{\pm 0.1}\uparrow$ |
| $\mathcal{L}_{\text{InfoNCE}}$ | ✓ | $0.1_{\pm 0.0}$ | $0.6_{\pm 0.1}$ | $1.1_{\pm 0.1}$ | $0.1_{\pm 0.0}$ | $0.5_{\pm 0.0}$ | $1.0_{\pm 0.0}$ | $3.4_{\pm 0.6}$ |
| $\mathcal{L}_{\text{InfoNCE+LTD}}, \eta = 0.05$ | ✓ | $\mathbf{24.7}_{\pm 0.5}\uparrow$ | $\mathbf{51.8}_{\pm 0.7}\uparrow$ | $\mathbf{65.6}_{\pm 1.4}\uparrow$ | $\mathbf{20.7}_{\pm 1.0}\uparrow$ | $\mathbf{49.2}_{\pm 0.6}\uparrow$ | $\mathbf{62.6}_{\pm 1.2}\uparrow$ | $\mathbf{274.6}_{\pm 4.6}\uparrow$ |
| | | | | MS-COCO | | | | |
| | | | | CLIP | | | | |
| $\mathcal{L}_{\text{InfoNCE}}$ | ✗ | $63.8_{\pm 0.3}$ | $86.1_{\pm 0.2}$ | $92.3_{\pm 0.0}$ | $46.3_{\pm 0.3}$ | $74.8_{\pm 0.1}$ | $84.1_{\pm 0.2}$ | $447.5_{\pm 0.5}$ |
| $\mathcal{L}_{\text{InfoNCE+LTD}}, \beta = 1$ | ✗ | $63.8_{\pm 0.0}-$ | $86.1_{\pm 0.0}-$ | $92.3_{\pm 0.0}-$ | $46.3_{\pm 0.0}-$ | $74.7_{\pm 0.0}-$ | $84.1_{\pm 0.0}-$ | $447.4_{\pm 0.0}-$ |
| $\mathcal{L}_{\text{InfoNCE}}$ | ✓ | $13.6_{\pm 0.9}$ | $31.5_{\pm 2.4}$ | $42.2_{\pm 3.7}$ | $7.3_{\pm 0.6}$ | $22.1_{\pm 1.0}$ | $32.7_{\pm 1.7}$ | $149.4_{\pm 32.7}$ |
| $\mathcal{L}_{\text{InfoNCE+LTD}}, \beta = 1$ | ✓ | $\mathbf{18.9}_{\pm 0.1}\uparrow$ | $\mathbf{41.8}_{\pm 0.1}\uparrow$ | $\mathbf{54.1}_{\pm 0.1}\uparrow$ | $\mathbf{16.5}_{\pm 0.0}\uparrow$ | $\mathbf{39.4}_{\pm 0.0}\uparrow$ | $\mathbf{52.6}_{\pm 0.1}\uparrow$ | $\mathbf{223.4}_{\pm 0.2}\uparrow$ |
| | | | | VSE++ | | | | |
| $\mathcal{L}_{\text{InfoNCE}}$ | ✗ | $42.2_{\pm 0.1}$ | $72.7_{\pm 0.1}$ | $83.2_{\pm 0.1}$ | $30.9_{\pm 0.0}$ | $61.2_{\pm 0.1}$ | $73.5_{\pm 0.1}$ | $363.8_{\pm 2.3}$ |
| $\mathcal{L}_{\text{InfoNCE+LTD}}, \eta = 0.1$ | ✗ | $\mathbf{43.6}_{\pm 0.1}\uparrow$ | $\mathbf{73.5}_{\pm 0.0}\uparrow$ | $\mathbf{83.7}_{\pm 0.0}\uparrow$ | $\mathbf{32.4}_{\pm 0.1}\uparrow$ | $\mathbf{62.5}_{\pm 0.0}\uparrow$ | $\mathbf{74.7}_{\pm 0.0}\uparrow$ | $\mathbf{370.5}_{\pm 0.1}\uparrow$ |
| $\mathcal{L}_{\text{InfoNCE}}$ | ✓ | $0.0_{\pm 0.0}$ | $0.1_{\pm 0.0}$ | $0.2_{\pm 0.0}$ | $0.0_{\pm 0.0}$ | $0.1_{\pm 0.0}$ | $0.2_{\pm 0.0}$ | $0.7_{\pm 0.0}$ |
| $\mathcal{L}_{\text{InfoNCE+LTD}}, \eta = 0.01$ | ✓ | $\mathbf{3.9}_{\pm 0.0}\uparrow$ | $\mathbf{13.7}_{\pm 0.6}\uparrow$ | $\mathbf{21.6}_{\pm 0.9}\uparrow$ | $\mathbf{3.1}_{\pm 0.2}\uparrow$ | $\mathbf{11.0}_{\pm 1.6}\uparrow$ | $\mathbf{18.1}_{\pm 3.0}\uparrow$ | $\mathbf{71.3}_{\pm 3.6}\uparrow$ |

Across the board, VSE++ models trained with the $\mathcal{L}_{\text{InfoNCE+LTD}}$ loss consistently outperform the $\mathcal{L}_{\text{InfoNCE}}$ loss, both for i2t and t2i retrieval and both when trained either with or without shortcuts, as indicated by higher recall@$k$ scores; this is consistent with the findings presented in (Bleeker et al., 2023)). For both the Flickr30k and MS-COCO dataset, when trained with the $\mathcal{L}_{\text{InfoNCE}}$ and with shortcuts present in the training data, the model performance collapses to around 0 in the absence of shortcuts (as we have seen in Section 4). However, when we train with shortcuts in the training data and with $\mathcal{L}_{\text{InfoNCE+LTD}}$, we observe, for both Flickr30k and MS-COCO, a significant gain in performance. The performance improvement is bigger for Flickr30k than for MS-COCO. In general, the recall scores are still significantly lower than training without shortcuts, however, the models do not solely rely on the shortcuts anymore to minimize the contrastive loss and are able during evaluation (in the absence of shortcuts) to still correctly match image-caption pairs with each other. The results in Table 1 show that LTD is able, in the presence of shortcuts in the training data, to guide (small-scale) VL models that are trained from scratch to not only learn the shortcut features that

Table 2: Mean and variance (over three training runs) recall@$k$ evaluation scores for the Flickr30k and MS-COCO datasets for image-to-text and text-to-image retrieval. We train with two loss functions: $\mathcal{L}_{\text{InfoNCE}}$ and $\mathcal{L}_{\text{InfoNCE+IFM}}$. We train either with (✓) or without (✗) shortcuts. For the model trained with $\mathcal{L}_{\text{InfoNCE+IFM}}$, we provide the hyper-parameters of the best-performing model.

| Loss | $S_{SynSC}$ | i2t | | | t2i | | | |
| --- | --- | --- | --- | --- | --- | --- | --- | --- |
| | | R@1 | R@5 | R@10 | R@1 | R@5 | R@10 | rsum |
| Flickr30k | | | | | | | | |
| CLIP | | | | | | | | |
| $\mathcal{L}_{\text{InfoNCE}}$ | ✗ | $86.9_{\pm 0.1}$ | $97.4_{\pm 0.0}$ | $98.8_{\pm 0.0}$ | $72.8_{\pm 0.2}$ | $92.1_{\pm 0.0}$ | $95.6_{\pm 0.0}$ | $543.5_{\pm 1.3}$ |
| $\mathcal{L}_{\text{InfoNCE+IFM}}, \epsilon = 0.05$ | ✗ | $\mathbf{87.4}_{\pm 0.1}\uparrow$ | $97.4_{\pm 0.2}-$ | $99.1_{\pm 0.0}-$ | $73.2_{\pm 0.0}-$ | $92.2_{\pm 0.0}-$ | $95.6_{\pm 0.0}-$ | $544.9_{\pm 0.2}-$ |
| $\mathcal{L}_{\text{InfoNCE}}$ | ✓ | $57.9_{\pm 0.3}$ | $84.6_{\pm 0.8}$ | $91.3_{\pm 0.0}$ | $43.9_{\pm 2.2}$ | $74.6_{\pm 0.8}$ | $84.4_{\pm 0.4}$ | $436.7_{\pm 18.8}$ |
| $\mathcal{L}_{\text{InfoNCE+IFM}}, \epsilon = 0.1$ | ✓ | $\mathbf{73.8}_{\pm 0.8}\uparrow$ | $\mathbf{91.5}_{\pm 0.5}\uparrow$ | $\mathbf{95.6}_{\pm 0.0}\uparrow$ | $\mathbf{58.9}_{\pm 0.1}\uparrow$ | $\mathbf{84.4}_{\pm 0.1}\uparrow$ | $\mathbf{91.1}_{\pm 0.2}\uparrow$ | $\mathbf{495.2}_{\pm 5.7}\uparrow$ |
| VSE++ | | | | | | | | |
| $\mathcal{L}_{\text{InfoNCE}}$ | ✗ | $\mathbf{52.9}_{\pm 0.2}$ | $\mathbf{80.5}_{\pm 0.1}$ | $\mathbf{87.6}_{\pm 0.4}$ | $\mathbf{40.5}_{\pm 0.1}$ | $68.8_{\pm 0.4}$ | $\mathbf{78.9}_{\pm 0.3}$ | $\mathbf{409.3}_{\pm 2.6}$ |
| $\mathcal{L}_{\text{InfoNCE+IFM}}, \epsilon = 0.05$ | ✗ | $52.4_{\pm 0.2}\downarrow$ | $76.9_{\pm 0.1}\downarrow$ | $85.3_{\pm 0.0}\downarrow$ | $39.1_{\pm 0.0}\downarrow$ | $68.8-_{\pm 0.1}$ | $78.2_{\pm 0.1}\downarrow$ | $400.7_{\pm 0.0}\downarrow$ |
| $\mathcal{L}_{\text{InfoNCE}}$ | ✓ | $0.1_{\pm 0.0}$ | $0.4_{\pm 0.0}$ | $0.8_{\pm 0.0}$ | $0.1_{\pm 0.0}$ | $0.4_{\pm 0.0}$ | $1.0_{\pm 0.0}$ | $2.9_{\pm 0.0}$ |
| $\mathcal{L}_{\text{InfoNCE+IFM}}, \epsilon = 0.05$ | ✓ | $0.0_{\pm 0.0}-$ | $0.6_{\pm 0.1}-$ | $0.9_{\pm 0.2}-$ | $0.1_{\pm 0.0}-$ | $0.5_{\pm 0.0}-$ | $1.0_{\pm 0.0}-$ | $3.2_{\pm 0.8}-$ |
| MS-COCO | | | | | | | | |
| CLIP | | | | | | | | |
| $\mathcal{L}_{\text{InfoNCE}}$ | ✗ | $\mathbf{63.5}_{\pm 0.1}$ | $\mathbf{86.0}_{\pm 0.3}$ | $\mathbf{92.2}_{\pm 0.0}$ | $46.3_{\pm 0.0}$ | $74.7_{\pm 0.0}$ | $84.2_{\pm 0.0}$ | $446.9_{\pm 0.9}$ |
| $\mathcal{L}_{\text{InfoNCE+IFM}}, \epsilon = 0.05$ | ✗ | $63.0_{\pm 0.1}\downarrow$ | $86.6_{\pm 0.1}\downarrow$ | $92.6_{\pm 0.2}\downarrow$ | $\mathbf{47.2}_{\pm 0.0}\uparrow$ | $\mathbf{75.6}_{\pm 0.0}\uparrow$ | $\mathbf{84.5}_{\pm 0.0}\uparrow$ | $\mathbf{449.5}_{\pm 1.7}\uparrow$ |
| $\mathcal{L}_{\text{InfoNCE}}$ | ✓ | $13.9_{\pm 0.0}$ | $32.7_{\pm 0.1}$ | $43.8_{\pm 0.0}$ | $8.8_{\pm 0.0}$ | $24.7_{\pm 0.2}$ | $35.5_{\pm 0.5}$ | $159.4_{\pm 3.4}$ |
| $\mathcal{L}_{\text{InfoNCE+IFM}}, \epsilon = 0.05$ | ✓ | $\mathbf{23.4}_{\pm 1.5}\uparrow$ | $\mathbf{46.5}_{\pm 2.7}\uparrow$ | $\mathbf{58.2}_{\pm 2.5}\uparrow$ | $\mathbf{17.1}_{\pm 0.3}\uparrow$ | $\mathbf{38.9}_{\pm 0.9}\uparrow$ | $\mathbf{51.3}_{\pm 1.0}\uparrow$ | $\mathbf{235.5}_{\pm 43.8}\uparrow$ |
| VSE++ | | | | | | | | |
| $\mathcal{L}_{\text{InfoNCE}}$ | ✗ | $\mathbf{41.7}_{\pm 0.3}$ | $\mathbf{72.5}_{\pm 0.1}$ | $\mathbf{83.1}_{\pm 0.1}$ | $\mathbf{31.3}_{\pm 0.0}$ | $61.1_{\pm 0.0}$ | $73.6_{\pm 0.0}$ | $\mathbf{363.4}_{\pm 0.4}$ |
| $\mathcal{L}_{\text{InfoNCE+IFM}}, \epsilon = 0.05$ | ✗ | $40.2_{\pm 0.0}\downarrow$ | $70.8_{\pm 0.1}\downarrow$ | $81.6_{\pm 0.1}\downarrow$ | $30.8_{\pm 0.0}\downarrow$ | $\mathbf{61.5}_{\pm 0.0}\uparrow$ | $\mathbf{74.3}_{\pm 0.0}\uparrow$ | $359.3_{\pm 1.1}\downarrow$ |
| $\mathcal{L}_{\text{InfoNCE}}$ | ✓ | $0.0_{\pm 0.0}$ | $0.1_{\pm 0.0}$ | $0.2_{\pm 0.0}$ | $0.0_{\pm 0.0}$ | $0.1_{\pm 0.0}$ | $0.2_{\pm 0.0}$ | $0.6_{\pm 0.0}$ |
| $\mathcal{L}_{\text{InfoNCE+IFM}}, \epsilon = 0.05$ | ✓ | $0.0_{\pm 0.0}-$ | $0.1_{\pm 0.0}-$ | $0.2_{\pm 0.0}-$ | $0.0_{\pm 0.0}-$ | $0.1_{\pm 0.0}-$ | $0.2_{\pm 0.0}-$ | $0.7_{\pm 0.0}-$ |

minimize the contrastive training objective but also represent other remaining task-relevant features in the data that are not extracted by $\mathcal{L}_{\text{InfoNCE}}$.

## 6.2 Does Implicit Feature Modification Reduce Shortcut Learning?

In Table 2 we summarize the effect of IFM on reducing shortcut solutions.

For CLIP, we observe that $\mathcal{L}_{\text{InfoNCE+IFM}}$, when training without shortcuts in the training data, only improves performance for the MS-COCO dataset for the t2i task. However, for both Flickr30k and MS-COCO we observe that, when training with unique shortcuts in the training data, fine-tuning with $\mathcal{L}_{\text{InfoNCE+IFM}}$ results in a significantly lower performance drop in recall score than when fine-tuning with the $\mathcal{L}_{\text{InfoNCE}}$. Similar to LTD, the recall@$k$ scores are still lower than when trained without shortcuts in the training data. We conclude that IFM is sufficient to reduce the suppression of features in favor of the shortcut features when fine-tuning a large-scale VL model, as indicated by higher recall@$k$ scores when evaluating without shortcuts.

For VSE++, both for the Flickr30k and MS-COCO dataset, we do not observe that $\mathcal{L}_{\text{InfoNCE+IFM}}$ outperforms the $\mathcal{L}_{\text{InfoNCE}}$, both with and without shortcuts present in the training data. We even observe that $\mathcal{L}_{\text{InfoNCE+IFM}}$, when training without shortcuts, results in a decrease in performance across all recall@$k$ metrics. When training with $\mathcal{L}_{\text{InfoNCE+IFM}}$ and with unique shortcuts in the training data, the evaluation performance still collapses to around 0. The results in Table 2 show that IFM is not sufficient to prevent models trained from scratch from fully collapsing to the artificial shortcut solutions we introduce in this work (as opposed to LTD).

## 6.3 Upshot

In this section, we have evaluated two methods for reducing shortcut learning on our SVL framework: LTD and IFM. LTD proves effective in reducing shortcut learning for both CLIP and VSE++. IFM demonstrates its efficacy solely during the fine-tuning of CLIP. These findings indicate that our SVL framework is a challenging and interesting framework to study and evaluate shortcut learning for contrastive VL models. Moreover, our results show that shortcut learning is only partially addressed by the evaluated methods since the evaluation results are not on par with the results on data lacking synthetic shortcuts.

## 7  Related work

We discuss related work on multi-view representation learning, vision-language learning, and shortcut learning.

**Multi-view Representation Learning.** To learn the underlying semantics of the training data, a subgroup of representation learning methods involves training neural encoders that maximize the agreement between representations of the similar *views* (van den Oord et al., 2018; Hjelm et al., 2019; Chen et al., 2020a; Radford et al., 2021; Bardes et al., 2022). In general, for uni-modal representation learning, data augmentations are used to generate different views of the same data point. One of the core assumptions in multi-view representation learning is that each view shares the same *task-relevant information* (Sridharan & Kakade, 2008; Zhao et al., 2017; Federici et al., 2020; Tian et al., 2020a; Shwartz-Ziv & LeCun, 2023). However, the optimal view for contrastive self-supervised learning (SSL) (i.e., which information is shared among views/which data augmentation is used) is task-dependent (Tian et al., 2020b; Xiao et al., 2021). Therefore, maximizing the mutual information (MI) between representations of views (i.e., shared information) does not necessarily result in representations that generalize better to down-stream evaluation tasks, since the representations may contain too much additional noise that is irrelevant for the downstream task (Tian et al., 2020b; Tschannen et al., 2020). An open problem in multi-view SSL is to learn representations that contain all task-relevant information from views where each view contains distinct, task-relevant information (Shwartz-Ziv & LeCun, 2023), this is especially a problem in the multi-modal learning domain (Zong et al., 2023).

Chen et al. (2021) investigate multi-view representation learning for images using contrastive losses. They demonstrate that when multiple competing features exist that redundantly predict the match between two views, contrastive models tend to focus on learning the easy-to-represent features while suppressing other task-relevant information. This results in contrastive losses mainly capturing the easy features, even if all task-relevant information is shared between the two views, suppressing the remaining relevant information.

Several optimization objectives have been introduced to either maximize the lower bound on the MI between views and their latent representations (van den Oord et al., 2018; Bachman et al., 2019; Hjelm et al., 2019; Tian et al., 2020a) or minimize the MI between representations of views while keeping the task-relevant information (Federici et al., 2020; Lee et al., 2021). To learn more task-relevant information that either might not be shared between views or that is compressed by a contrastive loss, several works proposed additional reconstruction objectives to maximize the MI between the latent representation and input data (Tsai et al., 2021; Wang et al., 2022; Li et al., 2023b; Bleeker et al., 2023). Liang et al. (2023) introduce a multimodal contrastive objective that factorizes the representations into shared and unique information, while also removing task-irrelevant information by minimizing the upper bound on MI between similar views.

**Vision-language Representation Learning.** The goal of VL representation learning is to combine information from the visual and textual modalities into a joint representation or learn coordinated represen-

tations (Baltrusaitis et al., 2019; Guo et al., 2019). The representation learning approaches can be separated into several groups.

*Contrastive methods* represent one prominent category of VL representation methods. The approaches in this group are typically dual encoders. Early methods in this category are trained from scratch; for instance, (Frome et al., 2013) proposed a VL representation learning model that features a skip-gram language model and a visual object categorization component trained with hinge rank loss. Another subgroup of methods uses a *dual-encoder* with a hinge-based triplet loss (Kiros et al., 2014; Li et al., 2019a; Lee et al., 2018). Kiros et al. (2014) use the loss for training a CNN-RNN dual encoder. Li et al. (2019a) leverage bottom-up attention and graph convolutional networks (Kipf & Welling, 2017) to learn the relationship between image regions. Lee et al. (2018) add stacked cross-attention to use both image regions and words as context.

More recently, contrastive approaches involve transformer-based dual-encoders trained with more data than the training data from the evaluation set(s). ALBEF (Li et al., 2021) propose to contrastively align unimodal representations before fusion, while X-VLM (Zeng et al., 2022) employs an additional cross-modal encoder to learn fine-grained VL representations. Florence (Yuan et al., 2021) leverages various adaptation models for learning fine-grained object-level representations. CLIP (Radford et al., 2021), a scaled-up dual-encoder, is pre-trained on the task of predicting which caption goes with which image. ALIGN (Jia et al., 2021) uses a simple dual-encoder trained on over a billion image alt-text pairs. FILIP (Yao et al., 2022) is a transformer-based bi-encoder that features late multimodal interaction meant to capture fine-grained representations. SLIP (Mu et al., 2022) combines language supervision and image self-supervision to learn visual representations without labels. DeCLIP (Li et al., 2022b) proposes to improve the efficiency of CLIP pretraining using intra-modality self-supervision, cross-modal multi-view supervision, and nearest neighbor supervision.

Another line of work includes learning VL representations using models that are inspired by BERT (Devlin et al., 2019). ViLBERT (Lu et al., 2019) and LXMERT (Tan & Bansal, 2019) expand upon BERT by introducing a two-stream architecture, where two transformers are applied to images and text independently, which is fused by a third transformer in a later stage. B2T2 (Alberti et al., 2019), VisualBERT (Li et al., 2019b), Unicoder-VL (Li et al., 2020a), VL-BERT (Su et al., 2020), and UNITER (Chen et al., 2020b) propose a single-stream architecture, where a single transformer is applied to both images and text. Oscar (Li et al., 2020b) uses caption object tags as anchor points that are fed to the transformer alongside region features. BEIT-3 (Wang et al., 2023) adapt multiway transformers trained using cross-entropy loss (Bao et al., 2022).

Another category of methods for learning VL representations are generative methods, that imply learning VL representation by generating new instances of one modality conditioned on the other modality. For instance, BLIP (Li et al., 2022a) bootstraps captions by generating synthetic captions and filtering out the noisy ones; BLIP-2 (Li et al., 2023a) bootstraps VL representation learning and, subsequently, vision-to-language generative learning. On the other hand, Tschannen et al. (2023) propose to pretrain a encoder-decoder architecture via the image captioning task.

**Shortcut Learning.** Geirhos et al. (2020) define shortcuts in deep neural networks as "decision rules that perform well on standard benchmarks but fail to transfer to more challenging testing conditions, such as real-world scenarios." In the context of deep learning, a shortcut solution can also be seen as a discrepancy between the features that a model has learned during training and the intended features that a model should learn to perform well during evaluation. For example, shortcuts might be features that minimize the training objective but are much easier to detect than the intended features that are relevant to the evaluation task. Shortcut learning can be caused by biases in the dataset or inductive biases in either the network architecture or training objective.

Hermann & Lampinen (2020) design a dataset with multiple predictive features, where each feature can be used as a label for an image classification task. The authors show that in the presence of multiple features that each redundantly predicts the target label, the deep neural model chooses to represent only one of the predictive features that are the easiest to detect, i.e., the model favors features that are easy to detect over features that are harder to discriminate. Next to that, they show that features that are not needed for a classification task, are in general suppressed by the model instead of captured in the learned latent representations.

Robinson et al. (2021) show that contrastive losses can have multiple local minima, where different local minima can be achieved by suppressing features from the input data (i.e., the model learns a shortcut by not learning all task-relevant features). To mitigate the shortcut learning problem, Robinson et al. (2021) propose implicit feature modification, a method that perpetuates the features of positive and negative samples during training to encourage the model to capture different features than the model currently relies on.

Scimeca et al. (2022) design an experimental set-up with multiple shortcut cues in the training data, where each shortcut is equally valid w.r.t. predicting the correct target label. The goal of the experimental setup is to investigate which cues are preferred to others when learning a classification task.

Latent target decoding (LTD) is a method to reduce predictive feature suppression (i.e., shortcuts) for resource-constrained contrastive ICR by reconstructing the input caption in a non-auto-regressive manner. Bleeker et al. (2023) argue that most of the task-relevant information for the ICR task is captured by the text modality. Hence, the focus is on the reconstruction of the text modality instead of the image modality. Bleeker et al. (2023) add a decoder to the learning algorithm, to reconstruct the input caption. Instead of reconstructing the input tokens, the input caption is reconstructed in a non-autoregressive manner in the latent space of a Sentence-BERT (Reimers & Gurevych, 2019; Song et al., 2020) model. LTD can be implemented as an optimization constraint and as a dual-loss. Li et al. (2023b) show that contrastive losses are prone to feature suppression. They introduce predictive contrastive learning (PCL), which combines contrastive learning with a decoder to reconstruct the input data from the latent representations to prevent shortcut learning.

Adnan et al. (2022) measure the MI between the latent representation and the input as a domain agnostic metric to find where (and when) in training a network relies on shortcuts in the input data. Their main finding is that, in the presence of shortcuts, the MI between the input data and the latent representation of the data is lower than without shortcuts in the input data. Hence, the latent representation captures less information of the input data in the presence of shortcuts and mainly relies on shortcuts to predict the target.

**Our Focus.** In this work, we focus on the problem of shortcut learning for VL in the context of multi-view VL representation learning with multiple captions per image. In contrast with previous (uni-modal) work on multi-view learning, we consider different captions matching to the same image as different *views.* We examine the problem by introducing a framework of synthetic shortcuts designed for VL representation learning, which allows us to investigate the problem in a controlled way. For our experiments, we select two prevalent VL models that are solely optimized with the InfoNCE loss: CLIP, a large-scale pre-trained model, and VSE++, a model trained from scratch. We select models that are solely optimized with a contrastive loss, to prevent measuring the effect of other optimization objectives on the shortcut learning problem.

## 8 Conclusion

In this work, we focus on the shortcut learning problem of contrastive learning in the context of vision-language (VL) representation learning with multiple captions per image. We have proposed synthetic shortcuts for vision-language (SVL): a training and evaluation framework to examine the problem of shortcut learning in a controlled way. The key component of this framework is synthetic shortcuts that we add to image-text data. Synthetic shortcuts represent additional, easily identifiable information that is shared between images and captions. We fine-tune CLIP and train a VSE++ model from scratch using our training framework to evaluate how prone contrastive VL models are to shortcut learning. Next, we have evaluated how shortcut learning can be partially mitigated using latent target decoding and implicit feature modification.

**Main Findings.** We have conducted experiments on two distinct VL models, CLIP and VSE++, and have evaluated the performance on Flickr30k and MS-COCO. We have found that when training with unique shortcuts, CLIP suppresses pre-trained features in favor of the shortcuts. VSE++ only learns to represent the shortcuts, when using unique shortcuts, showing that none of the remaining task-relevant (both shared and unique) information is captured by the encoders when training a model from scratch. When using *n bits of shortcuts*, we have shown that the more bits we use, the more the contrastive VL models rely on the synthetic shortcuts. Our results demonstrate that contrastive VL methods tend to depend on easy-to-learn discriminatory features shared among images and all matching captions while suppressing the

remaining task-relevant information. Next, we have evaluated two methods for reducing shortcut learning on our framework of synthetic shortcuts for image-caption datasets. Both methods partially mitigate shortcut learning when training and evaluating with our shortcut learning framework. These findings show that our framework is a challenging framework to study and evaluate shortcut learning for contrastive VL and underline the complexity of our framework in studying and evaluating shortcut learning within the context of contrastive VL representation learning.

**Implications.** The implications of our findings are twofold. First, we examine the limitations of contrastive optimization objectives for VL representation learning, demonstrating that they predominantly capture features that are easily discriminable but may not necessarily constitute task-optimal representations. Second, our work contributes a novel framework for investigating shortcut learning problem in the context of VL representation learning with multiple captions per image, providing insights into the extent to which models rely on shortcuts when they are available and how existing shortcut reduction methods are capable of reducing shortcut learning when training with our framework.

**Limitations.** Some of the limitations of our work are related to the fact that we focused on two specific models, one optimization objective (InfoNCE), and two datasets, and the generalizability of our findings to other VL models, optimization objectives, and datasets warrants further exploration. Additionally, the synthetic shortcuts introduced in this work are not dependent on image-caption pairs. Our training and evaluation setup shows that, in the presence of shortcuts in the training data, contrastive VL models mainly rely on the easy-to-detect shortcut features, which indicates that the InfoNCE loss cannot learn tasks-optimal representations for VL tasks when multiple captions are used for training. However, it remains unclear to what degree the unique information of the captions is captured by the contrastive loss VL models.

**Future Work.** We suggest working on the development of optimization objectives that specifically address the shortcut learning problem for VL training with multiple captions per image. We also suggest extending our synthetic shortcuts for image-caption datasets to a framework with unique shortcut information per caption. By having unique shortcut information per caption, it becomes possible to measure how much of the shared/caption-specific shortcut information is captured by encoder models. Another future direction includes investigating alternative training strategies or loss functions to further mitigate shortcut learning problems. Another promising direction for future work includes the improvement of existing methods or the exploration of novel techniques that address the limitations of existing shortcut reduction methods, potentially through the combination of multiple approaches. Extending the SVL framework to better capture nuances and complexities of natural data is another important direction that would facilitate a more comprehensive understanding of the implications of shortcut learning in real-world scenarios and datasets.

## 9 Broader Impact

This paper motivates and introduces a framework for investigating the problem of shortcut learning for contrastive VL representation learning with multiple captions per image in a controlled way. It also examines how two shortcut learning reduction methods perform on the proposed framework. Overall, the framework provides a tool for analyzing and understanding the problem of shortcut learning in the context of contrastive VL representation learning; it can be used in various settings that require deeper insight into the quality of learned VL representations.

We should be aware that the reliance on shortcuts in VLMs poses ethical concerns with potential real-world implications. Models that learn shortcuts may overlook nuanced details in images and text, leading to biased or inaccurate outcomes. Furthermore, the transparency and explainability of VLMs are crucial considerations. Models that rely on shortcuts may make decisions based on features that are not easily interpretable or explainable to users. This lack of transparency can diminish trust in AI systems.

## Acknowledgements

We thank Marco Federici and Mathijs Henquet for the discussions on mutual information and feedback on the draft. Additionally, we thank Shashank Gupta and Panagiotis Efstratiadis for helpful feedback.

This research was supported by the Nationale Politie, Ahold Delhaize, project IDEAS with project number VI.Vidi.223.166 of the NWO Talent Programme, which is (partly) financed by the Dutch Research Council (NWO), the Hybrid Intelligence Center, a 10-year program funded by the Dutch Ministry of Education, Culture and Science through the Netherlands Organisation for Scientific Research, https://hybrid-intelligence-centre.nl, project LESSEN with project number NWA.1389.20.183 of the research program NWA ORC 2020/21, which is (partly) financed by the Dutch Research Council (NWO), project ROBUST with project number KICH3.LTP.20.006, which is (partly) financed by the Dutch Research Council (NWO), DPG Media, RTL, and the Dutch Ministry of Economic Affairs and Climate Policy (EZK) under the program LTP KIC 2020-2023, and the FINDHR (Fairness and Intersectional Non-Discrimination in Human Recommendation) project that received funding from the European Union's Horizon Europe research and innovation program under grant agreement No 101070212. All content represents the opinion of the authors, which is not necessarily shared or endorsed by their respective employers and/or sponsors.

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

## A   Notation

Table 3: Notation used in the paper.

| Symbol | Description |
|---|---|
| $\mathcal{L}_{\text{InfoNCE}}$ | InfoNCE loss |
| $\mathcal{L}_{\text{InfoNCE+LTD}}$ | Loss that combines InfoNCE and latent target decoding (LTD) |
| $\mathcal{L}_{\text{InfoNCE+IFM}}$ | Loss that combines InfoNCE and implicit feature modification (IFM) |
| $\mathcal{L}_{recon}$ | Reconstruction loss |
| $\mathcal{D}$ | Dataset $\mathcal{D}$ that comprises $N$ image-caption tuples: $\mathcal{D} = \left\{ \left( \mathbf{x}_{\mathcal{I}}^i, \{\mathbf{x}_{\mathcal{C}_j}^i\}_{j=1}^k \right) \right\}_{i=1}^N$; $i$-th image-caption tuple in the dataset $\mathcal{D}$ consist out of an image $\mathbf{x}_{\mathcal{I}}^i$ and $k$ associated captions $\{\mathbf{x}_{\mathcal{C}_j}^i\}_{j=1}^k$ |
| $\mathcal{B}$ | Batch of image-caption pairs |
| $\mathbf{x}_{\mathcal{I}}$ | Image |
| $\mathbf{x}_{\mathcal{C}}$ | Caption |
| $\mathbf{z}_{\mathcal{I}}$ | Latent representation of image $\mathbf{x}_{\mathcal{I}}$ |
| $\mathbf{z}_{\mathcal{C}}$ | Latent representation of caption $\mathbf{x}_{\mathcal{C}}$ |
| $\mathbf{z}_{\mathcal{C}\rightarrow\mathcal{I}}^{SUF}$ | Latent representation of caption $\mathbf{x}_{\mathcal{C}}$ that is sufficient for image $\mathbf{x}_{\mathcal{I}}$ |
| $\mathbf{z}_{\mathcal{I}\rightarrow\mathcal{C}}^{SUF}$ | Latent representation of image $\mathbf{x}_{\mathcal{I}}$ sufficient for caption $\mathbf{x}_{\mathcal{C}}$ |
| $\mathbf{z}_{\mathcal{I}\rightarrow\mathcal{C}}^{MIN}$ | Latent representation of image $\mathbf{x}_{\mathcal{I}}$ that is minimal sufficient for caption $\mathbf{x}_{\mathcal{C}}$ |
| $\mathbf{z}_{\mathcal{I}\rightarrow K}^{OPT}$ | Latent representation of image $\mathbf{x}_{\mathcal{I}}$ that is optimal for set of captions $K$ given task $T$ |
| $R$ | Task-relevant information |
| $\neg R$ | Task-irrelevant information |
| $C$ | Task-relevant information specific for a caption $\mathbf{x}_{\mathcal{C}}$ |
| $S_{SynSC}$ | Synthetic shortcut |
| $S$ | Original shared information |
| $S^+$ | Shared information that includes synthetic shortcut |
| $R^+$ | Task-relevant information that contains synthetic shortcut |
| $f_\theta(\cdot)$ | Image encoder parametrised by $\theta$; takes image $\mathbf{x}_{\mathcal{I}}$ as input and returns its latent representation $\mathbf{z}_{\mathcal{I}}$: $\mathbf{z}_{\mathcal{I}} := f_\theta(\mathbf{x}_{\mathcal{I}})$ |
| $g_\phi(\cdot)$ | Caption encoder parametrised by $\phi$; takes caption $\mathbf{x}_{\mathcal{C}}$ as input and returns its latent representation $\mathbf{z}_{\mathcal{C}}$: $\mathbf{z}_{\mathcal{C}} := g_\phi(\mathbf{z}_{\mathcal{C}})$ |
| $\tau$ | Temperature paramater of $\mathcal{L}_{\text{InfoNCE}}$ |
| $\epsilon$ | Perturbation budget for $\mathcal{L}_{\text{IFM}}$ |
| $\eta$ | Reconstruction bound for $\mathcal{L}_{\text{LTD}}$ |

## B   Problem Definition and Assumptions

In this work, we solely focus on contrastive VL representation learning. We work in a setting where we investigate the problem by fine-tuning a large pre-trained foundation model (CLIP, Radford et al., 2021) and training a resource-constrained image-text method from scratch (VSE++, Faghri et al., 2018). We train and

evaluate using two benchmark datasets where multiple captions per image are available: Flickr30k (Young et al., 2014) and MS-COCO Captions (Lin et al., 2014). Both datasets come with 5 captions per image. We work in a dual-encoder setup, i.e., we have a separate image and caption encoder, which do not share parameters.

## B.1 Evaluation Task

The image-caption retrieval (ICR) evaluation task, consists of two sub-tasks: image-to-text (i2t) and text-to-image (t2i) retrieval. In ICR, either an image or a caption is used as a query and the goal is to rank a set of candidates in the other modality. In this work, we follow the standard ICR evaluation procedure (see, e.g., Faghri et al., 2018; Lee et al., 2018; Li et al., 2019a). The evaluation metric for the ICR task is Recall@$k$, with $k = \{1, 5, 10\}$. For t2i retrieval, there is one matching/positive image per query caption (when using the Flickr30k or MS-COCO or dataset). Hence, the Recall@$k$ metric represents how often the correct image is present in the top-$k$ of the ranking. For i2t retrieval, however, there are 5 matching captions per image. Therefore, only the highest-ranked correct caption is taken into account when measuring the Recall@$k$ (i.e., in the highest-ranked caption present in the top $k$). Standard practice to select the best model checkpoint during training is to use the *recall sum* (rsum) as a validation metric. The recall sum is the sum of recall at 1, 5, and 10, for both i2t and t2i. Therefore, the maximum value of the recall sum is 600.

## B.2 Assumptions

Throughout this work, we rely on several assumptions about the problem definition. Our assumptions are defined at the level of an image-text tuple. Following Section 2, we formalize the assumptions on the case where one image is associated with two captions: $\left(\mathbf{x}_{\mathcal{I}}, \{\mathbf{x}_{\mathcal{C}_A}, \mathbf{x}_{\mathcal{C}_B}\}\right)$.

**Assumption 1.** *Each caption in the tuple contain information that is distinct from the other captions in the tuple and all captions and image in the tuple contain shared and unique information:*

$$I(\mathbf{x}_{\mathcal{I}}; \mathbf{x}_{\mathcal{C}_A}; \mathbf{x}_{\mathcal{C}_B}) > 0$$
$$I(\mathbf{x}_{\mathcal{I}}; \mathbf{x}_{\mathcal{C}_A} \mid \mathbf{x}_{\mathcal{C}_B}) > 0, \ I(\mathbf{x}_{\mathcal{I}}; \mathbf{x}_{\mathcal{C}_B} \mid \mathbf{x}_{\mathcal{C}_A}) > 0 \ and \ I(\mathbf{x}_{\mathcal{C}_A}; \mathbf{x}_{\mathcal{C}_B} \mid \mathbf{x}_{\mathcal{I}}) > 0$$
$$H(\mathbf{x}_{\mathcal{I}} \mid \mathbf{x}_{\mathcal{C}_A}, \mathbf{x}_{\mathcal{C}_B}) > 0, \ H(\mathbf{x}_{\mathcal{C}_A} \mid \mathbf{x}_{\mathcal{I}}, \mathbf{x}_{\mathcal{C}_B}) > 0 \ and \ H(\mathbf{x}_{\mathcal{C}_B} \mid \mathbf{x}_{\mathcal{I}}, \mathbf{x}_{\mathcal{C}_A}) > 0.$$

**Assumption 2.** *Task-relevant information $R$ is the combination of all the information shared between an image and each caption in the tuple:*

$$R = I(\mathbf{x}_{\mathcal{I}}; \mathbf{x}_{\mathcal{C}_A} \mid \mathbf{x}_{\mathcal{C}_B}) + I(\mathbf{x}_{\mathcal{I}}; \mathbf{x}_{\mathcal{C}_B} \mid \mathbf{x}_{\mathcal{C}_A}) + I(\mathbf{x}_{\mathcal{I}}; \mathbf{x}_{\mathcal{C}_A}; \mathbf{x}_{\mathcal{C}_B}).$$

## C   Analysis of Contrastive Learning for Multiple Captions per Image

**Theorem 1** (Suboptimality of contrastive learning with multiple captions per image)**.** *Given an image $\mathbf{x}_{\mathcal{I}}$, a set of matching captions $\mathcal{C} = \{\mathbf{x}_{\mathcal{C}_A}, \mathbf{x}_{\mathcal{C}_B}\}$, and a contrastive learning loss function $\mathcal{L}_{InfoNCE}$ that optimizes for task $T$, image representations learned during contrastive learning will be minimal sufficient and will never be task-optimal image representations. More formally, assume that:*

$(H_1)$ $\forall i, j \in \{A, B\}$ *such that* $i \neq j$, $I(\mathbf{z}_{\mathcal{I} \to \mathcal{C}_i}^{MIN}; \mathbf{x}_{\mathcal{C}_i}) = I(\mathbf{x}_{\mathcal{I}}; \mathbf{x}_{\mathcal{C}_i} \mid \mathbf{x}_{\mathcal{C}_j}) + I(\mathbf{x}_{\mathcal{I}}; \mathbf{x}_{\mathcal{C}_i}; \mathbf{x}_{\mathcal{C}_j}).$

$(H_2)$ $\exists i, j \in \{A, B\}$ *with* $i \neq j$ *such that* $I(\mathbf{x}_{\mathcal{I}}; \mathbf{x}_{\mathcal{C}_i} \mid \mathbf{x}_{\mathcal{C}_j}) > 0.$

*Then the following holds:*

$(T_2)$ $\exists \ i \in \{A, B\}$ *such that* $I(\mathbf{z}_{\mathcal{I} \to \mathcal{C}}^{OPT}; \mathbf{x}_{\mathcal{C}_A} \mathbf{x}_{\mathcal{C}_B}) > I(\mathbf{z}_{\mathcal{I} \to \mathcal{C}_i}^{MIN}; \mathbf{x}_{\mathcal{C}_i}).$

*Proof.* Following Eq. 1 we define a task-optimal representation of an image $\mathbf{x}_{\mathcal{I}}$ w.r.t. all matching captions in $\mathcal{C}$ as:

$$I(\mathbf{z}_{\mathcal{I} \to \mathcal{C}}^{OPT}; \mathbf{x}_{\mathcal{C}_A} \mathbf{x}_{\mathcal{C}_B}) = \underbrace{I(\mathbf{x}_{\mathcal{I}}; \mathbf{x}_{\mathcal{C}_A} \mid \mathbf{x}_{\mathcal{C}_B})}_{C_A} + \underbrace{I(\mathbf{x}_{\mathcal{I}}; \mathbf{x}_{\mathcal{C}_B} \mid \mathbf{x}_{\mathcal{C}_A})}_{C_B} + \underbrace{I(\mathbf{x}_{\mathcal{I}}; \mathbf{x}_{\mathcal{C}_A}; \mathbf{x}_{\mathcal{C}_B})}_{S}.$$

Furthermore, following Definition 2.3, we define minimal sufficient representations of image $\mathbf{x}_\mathcal{I}$ w.r.t. each matching caption in $\mathcal{C}$ as a combination of caption-specific and shared information:

$$I(\mathbf{z}_{\mathcal{I}\to\mathcal{C}_A}^{MIN}; \mathbf{x}_{\mathcal{C}_A}) = \underbrace{I(\mathbf{x}_\mathcal{I}; \mathbf{x}_{\mathcal{C}_A} \mid \mathbf{x}_{\mathcal{C}_B})}_{C_A} + \underbrace{I(\mathbf{x}_\mathcal{I}; \mathbf{x}_{\mathcal{C}_A}; \mathbf{x}_{\mathcal{C}_B})}_{S}$$

$$I(\mathbf{z}_{\mathcal{I}\to\mathcal{C}_B}^{MIN}; \mathbf{x}_{\mathcal{C}_B}) = \underbrace{I(\mathbf{x}_\mathcal{I}; \mathbf{x}_{\mathcal{C}_B} \mid \mathbf{x}_{\mathcal{C}_A})}_{C_B} + \underbrace{I(\mathbf{x}_\mathcal{I}; \mathbf{x}_{\mathcal{C}_A}; \mathbf{x}_{\mathcal{C}_B})}_{S}.$$

Following assumption $H_2$, for at least one caption $\mathbf{x}_\mathcal{C} \in \mathcal{C}$ associated with the image $\mathbf{x}_\mathcal{I}$, caption-specific information is positive. Therefore, we consider two cases:

- If caption-specific information of $\mathbf{x}_{\mathcal{C}_A}$ is positive, that is, if $I(\mathbf{x}_\mathcal{I}; \mathbf{x}_{\mathcal{C}_A} \mid \mathbf{x}_{\mathcal{C}_B}) > 0$:

$$\underbrace{I(\mathbf{x}_\mathcal{I}; \mathbf{x}_{\mathcal{C}_A} \mid \mathbf{x}_{\mathcal{C}_B}) + I(\mathbf{x}_\mathcal{I}; \mathbf{x}_{\mathcal{C}_B} \mid \mathbf{x}_{\mathcal{C}_A}) + I(\mathbf{x}_\mathcal{I}; \mathbf{x}_{\mathcal{C}_A}; \mathbf{x}_{\mathcal{C}_B})}_{(\mathbf{z}_{\mathcal{I}\to\mathcal{C}}^{OPT}; \mathbf{x}_{\mathcal{C}_A}\mathbf{x}_{\mathcal{C}_B})} > \underbrace{I(\mathbf{x}_\mathcal{I}; \mathbf{x}_{\mathcal{C}_B} \mid \mathbf{x}_{\mathcal{C}_A}) + I(\mathbf{x}_\mathcal{I}; \mathbf{x}_{\mathcal{C}_A}; \mathbf{x}_{\mathcal{C}_B})}_{I(\mathbf{z}_{\mathcal{I}\to\mathcal{C}_B}^{MIN}; \mathbf{x}_{\mathcal{C}_B})} \Rightarrow$$

$$\Rightarrow I(\mathbf{z}_{\mathcal{I}\to\mathcal{C}}^{OPT}; \mathbf{x}_{\mathcal{C}_A}\mathbf{x}_{\mathcal{C}_B}) > I(\mathbf{z}_{\mathcal{I}\to\mathcal{C}_B}^{MIN}; \mathbf{x}_{\mathcal{C}_B}).$$

- Similarly, if caption-specific information of $\mathbf{x}_{\mathcal{C}_B}$ is positive, that is, if $I(\mathbf{x}_\mathcal{I}; \mathbf{x}_{\mathcal{C}_B} \mid \mathbf{x}_{\mathcal{C}_A}) > 0$:

$$\underbrace{I(\mathbf{x}_\mathcal{I}; \mathbf{x}_{\mathcal{C}_A} \mid \mathbf{x}_{\mathcal{C}_B}) + I(\mathbf{x}_\mathcal{I}; \mathbf{x}_{\mathcal{C}_B} \mid \mathbf{x}_{\mathcal{C}_A}) + I(\mathbf{x}_\mathcal{I}; \mathbf{x}_{\mathcal{C}_A}; \mathbf{x}_{\mathcal{C}_B})}_{(\mathbf{z}_{\mathcal{I}\to\mathcal{C}}^{OPT}; \mathbf{x}_{\mathcal{C}_A}\mathbf{x}_{\mathcal{C}_B})} > \underbrace{I(\mathbf{x}_\mathcal{I}; \mathbf{x}_{\mathcal{C}_A} \mid \mathbf{x}_{\mathcal{C}_B}) + I(\mathbf{x}_\mathcal{I}; \mathbf{x}_{\mathcal{C}_A}; \mathbf{x}_{\mathcal{C}_B})}_{I(\mathbf{z}_{\mathcal{I}\to\mathcal{C}_A}^{MIN}; \mathbf{x}_{\mathcal{C}_A})} \Rightarrow$$

$$\Rightarrow I(\mathbf{z}_{\mathcal{I}\to\mathcal{C}}^{OPT}; \mathbf{x}_{\mathcal{C}_A}\mathbf{x}_{\mathcal{C}_B}) > I(\mathbf{z}_{\mathcal{I}\to\mathcal{C}_A}^{MIN}; \mathbf{x}_{\mathcal{C}_A}).$$

Therefore, we show that in a setup where a single image is associated with multiple captions, and given at least one caption contains caption-specific information, image representations learned contrastively w.r.t. associated captions would contain less information than task-optimal image representation: $\exists\, i \in \{A, B\}$ such that $I(\mathbf{z}_{\mathcal{I}\to\mathcal{C}}^{OPT}; \mathbf{x}_{\mathcal{C}_A}\mathbf{x}_{\mathcal{C}_B}) > I(\mathbf{z}_{\mathcal{I}\to\mathcal{C}_i}^{MIN}; \mathbf{x}_{\mathcal{C}_i})$. $\qquad\square$

## D  Experimental Setup

### D.1  Datasets

**Flickr30k** consists of 31,000 images annotated with 5 matching captions (Young et al., 2014).

**MS-COCO** consists of 123,287 images, each image annotated with 5 matching captions (Lin et al., 2014). The original dataset was introduced for large-scale object recognition.

For both datasets, we use the training, validation, and test splits from (Karpathy & Li, 2015).

### D.2  Models

We use CLIP and VSE++. Both consist of an image and a text encoder that do not share parameters.

**CLIP** is a large-scale image-text foundation model (Radford et al., 2021). The model is pre-trained on a collection of 400 million image-text pairs collected from the Web. The encoders are pre-trained using a contrastive loss (InfoNCE) on image-text pairs. The text encoder of consists of a 12-layer transformer model, described in (Radford et al., 2019). As for the image encoder, CLIP utilizes various model backbones, such as ResNet (He et al., 2016) and Vision Transformer (Dosovitskiy et al., 2021). In this work, we use the ResNet-50 ('RN50') variant of the CLIP image encoder.[2] The CLIP encoders are trained to jointly understand images

---

[2] https://github.com/openai/CLIP/

and text. Therefore, the learned representations generalize to a wide range of different zero-shot (visual) evaluation tasks, such as classification, without task-specific fine-tuning, by using textual prompts.

**VSE++** is an image-caption encoder trained from scratch (Faghri et al., 2018). The model features a triplet loss function with a margin parameter $\alpha = 0.2$. The text encoder is a one-layer gated recurrent unit (GRU) (Cho et al., 2014). The available image encoder configurations are ResNet-152 (He et al., 2016) and VGG19 (Simonyan & Zisserman, 2015). In this work, we use ResNet-152.

### D.3 Training

**CLIP.** To fine-tune CLIP, we follow (Yuksekgonul et al., 2023). All models are fine-tuned for 5 epochs. We employ a cosine-annealing learning rate schedule, with a base learning rate of $2e-5$, and 100 steps of warm-up. As an optimizer, we use AdamW (Loshchilov & Hutter, 2019) with a gradient clipping value of 2. For the InfoNCE loss, we use the logit-scale (i.e., temperature $\tau$) from the pre-trained CLIP model and fine-tune the logit-scale end-to-end along with the rest of the model parameters.

**VSE++.** The model is trained for 30 epochs using a linear learning rate schedule with a base learning rate of $2e-4$. We use the Adam optimizer (Kingma & Ba, 2015) with a gradient clipping value of 2. Instead of the triplet loss, we use the InfoNCE loss similar to Radford et al. (2021),

For both models, instead of selecting the best-performing model based on the validation set scores, we use the final checkpoint at the end of training.

### D.4 Shortcut Sampling

Our goal is to add the shortcuts in a manner that preserves the original information of the images and captions. For the captions, we append the shortcut at the end of the captions. In order to prevent a tokenizer from tokenizing the shortcut into a single token, we insert spaces between each number of the shortcut. For the images, we place the numbers of the shortcuts at the top of the images, evenly spaced across the entire width of the images (to make sure the shortcut is evenly spaced across the feature map of the image). We always use 6 digits to represent a shortcut. If a shortcut number contains fewer than 6 digits, we fill the remaining positions with zeros for padding. For the MNIST images, we always sample a random image from the set of images representing the number that belongs to (also during evaluation), to prevent overfitting on specific MNIST images. In Figure 6, we provide four examples of image-caption pairs with randomly added shortcuts. The examples in Figure 6 show (i) how synthetic shortcuts are added to the image and the caption, and (ii) that the shortcuts preserve the original (task-relevant) information of the images and captions.

## E   Optimization Objectives

### E.1   InfoNCE

In this work, we use InfoNCE loss, $\mathcal{L}_{\text{InfoNCE}}$ (van den Oord et al., 2018). Given a dual-encoder setup, we optimize a model in two directions: image-to-text (i2t) and text-to-image (t2i). The loss is defined as follows:

$$\mathcal{L}_{\text{InfoNCE}}^{i2t} = \frac{1}{|\mathcal{B}|} \sum_{i \in \mathcal{B}} \log \frac{\exp(\mathbf{z}_{\mathcal{I}}^i \mathbf{z}_{\mathcal{C}}^i / \tau)}{\exp(\mathbf{z}_{\mathcal{I}}^i \mathbf{z}_{\mathcal{C}}^i / \tau) + \sum_{j \neq i} \exp(\mathbf{z}_{\mathcal{I}}^i \mathbf{z}_{\mathcal{C}}^j / \tau)},$$

$$\mathcal{L}_{\text{InfoNCE}}^{i2t} = \frac{1}{|\mathcal{B}|} \sum_{i \in \mathcal{B}} \log \frac{\exp(\mathbf{z}_{\mathcal{I}}^i \mathbf{z}_{\mathcal{C}}^i / \tau)}{\exp(\mathbf{z}_{\mathcal{I}}^i \mathbf{z}_{\mathcal{C}}^i / \tau) + \sum_{j \neq i} \exp(\mathbf{z}_{\mathcal{I}}^j \mathbf{z}_{\mathcal{C}}^i / \tau)},$$

$$\mathcal{L}_{\text{InfoNCE}} = \frac{1}{2} \mathcal{L}_{\text{InfoNCE}}^{i2t} + \frac{1}{2} \mathcal{L}_{\text{InfoNCE}}^{t2i}.$$

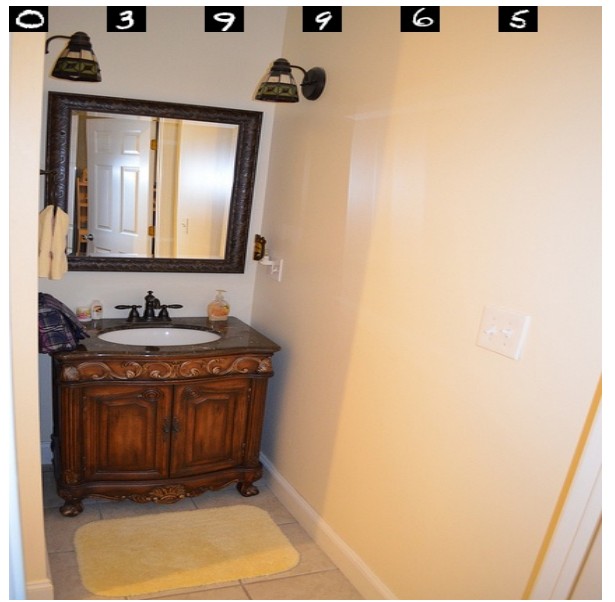

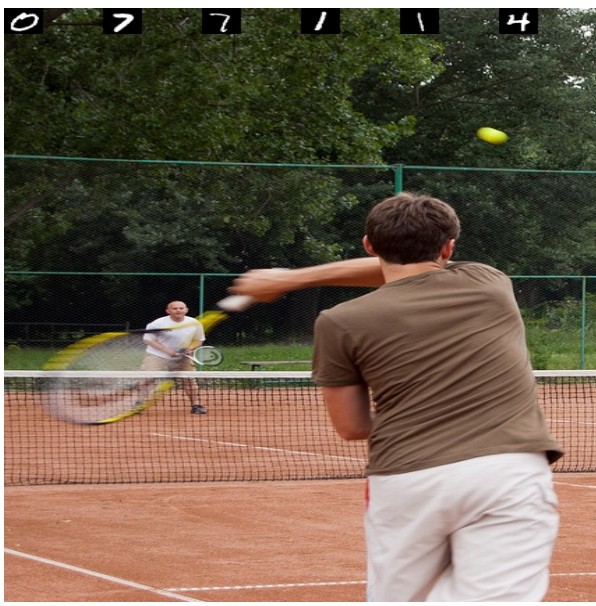

(a) **Caption**: "A bathroom sink with wood finish cabinets. 0 3 9 9 6 5."

(b) **Caption**: "A guy in a brown shirt has just hit a tennis ball. 0 7 7 1 1 4."

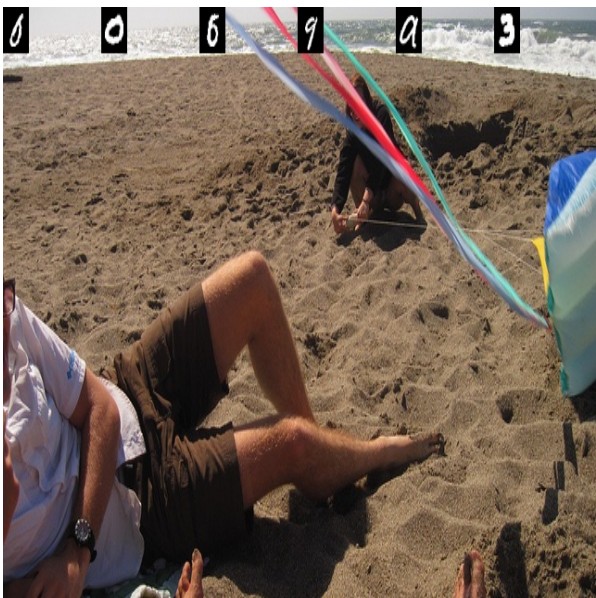

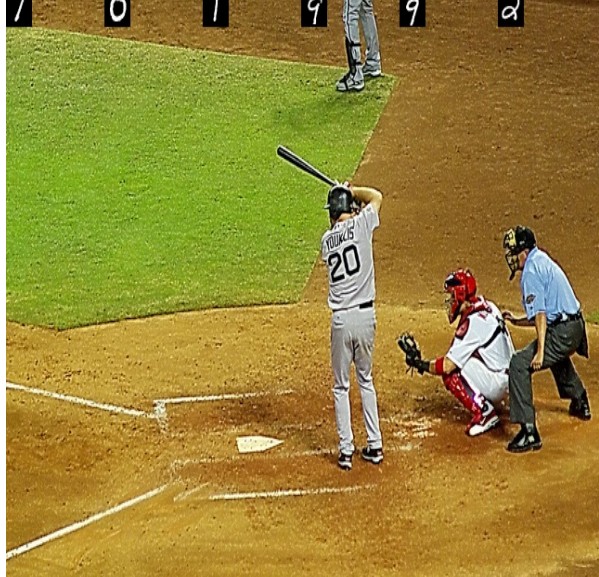

(c) **Caption**: "A man in shorts is lying on the beach. 0 0 6 9 9 3."

(d) **Caption**: "A player up to bat in a baseball game. 1 0 1 9 9 2."

Figure 6: Four random samples from the MS-COCO dataset including shortcuts added on both the image and caption.

### E.2  Latent Target Decoding

Latent target decoding (LTD) (Bleeker et al., 2023) is an optimization objective that reduces predictive feature suppression for resource-constrained VL methods. LTD consists of $\mathcal{L}_{\text{InfoNCE}}$ and a reconstruction loss $\mathcal{L}_{recon}$, which reconstructs the input caption from the latent representation $\mathbf{z}_{\mathcal{C}}$.

In (Bleeker et al., 2023), LTD is implemented in two ways. Firstly, as a dual optimization objective:

$$\mathcal{L}_{\text{InfoNCE+LTD}} = \mathcal{L}_{\text{InfoNCE}} + \beta \mathcal{L}_{recon}.$$

Secondly, as an optimization constraint in combination with gradient descent by using the method of Lagrange multipliers:

$$\max_{\lambda} \min \mathcal{L}_{\text{InfoNCE+LTD}} = \mathcal{L}_{\text{InfoNCE}} + \lambda \left( \frac{\mathcal{L}_{recon}}{\eta} - 1 \right).$$

This optimization objective is minimized w.r.t. model parameter, while also being maximized w.r.t. $\lambda$. The value of $\lambda$ is automatically tuned by gradient ascent, such that the reconstruction bound $\eta$ is met. In this work, we use both LTD as a dual optimization objective and an optimization constraint. We select the loss with the highest evaluation scores on the validation set for evaluation.

### E.3 Implicit Feature Modification

Implicit feature modification (IFM) (Robinson et al., 2021) is a contrastive loss, with an additional perturbation budget $\epsilon$. IFM perturbs the logits value of the similarity scores between the images and captions, such that the model avoids using shortcut solutions for a correct similarity score. IFM subtracts $\epsilon/\tau$ from the positive logit values and adds $\epsilon/\tau$ to the negative logits values.

$$\mathcal{L}_{\text{IFM}}^{t2i} = \frac{1}{|\mathcal{B}|} \sum_{i \in \mathcal{B}} \log \frac{\exp((\mathbf{z}_{\mathcal{I}}^i \mathbf{z}_{\mathcal{C}}^i - \epsilon)/\tau)}{\exp((\mathbf{z}_{\mathcal{I}}^i \mathbf{z}_{\mathcal{C}}^i) - \epsilon)/\tau) + \sum_{j \neq i} \exp((\mathbf{z}_{\mathcal{I}}^i \mathbf{z}_{\mathcal{C}}^j + \epsilon)/\tau)},$$

$$\mathcal{L}_{\text{IFM}}^{i2t} = \frac{1}{|\mathcal{B}|} \sum_{i \in \mathcal{B}} \log \frac{\exp((\mathbf{z}_{\mathcal{I}}^i \mathbf{z}_{\mathcal{C}}^i - \epsilon)/\tau)}{\exp((\mathbf{z}_{\mathcal{I}}^i \mathbf{z}_{\mathcal{C}}^i) - \epsilon)/\tau) + \sum_{j \neq i} \exp((\mathbf{z}_{\mathcal{I}}^j \mathbf{z}_{\mathcal{C}}^i + \epsilon)/\tau)},$$

$$\mathcal{L}_{\text{IFM}} = \frac{1}{2} \mathcal{L}_{\text{IFM}}^{t2i} + \frac{1}{2} \mathcal{L}_{\text{IFM}}^{i2t},$$

$$\mathcal{L}_{\text{InfoNCE+IFM}} = \frac{1}{2} \mathcal{L}_{\text{IFM}} + \frac{1}{2} \mathcal{L}_{\text{InfoNCE}}.$$

Similar to Robinson et al. (2021), we combine IFM and the InfoNCE in a dual optimization objective.

