# OpenReview forum: "Demonstrating and Reducing Shortcuts in Vision-Language Representation Learning"
_TMLR — Accepted by TMLR_

### Review · Reviewer_RZvX · 2024-03-26

**Summary Of Contributions:**

The paper studies the problem of contrastive learning for vision-language models with multiple captions. The paper first shows that, in theory, InfoNCE loss could not capture the whole task information in multiple captions and will be prone to shortcut features in the data. The paper then proposes a way to introduce shortcut information into the dataset and shows that the contrastive learning performance degrades terribly. The paper then evaluates two existing methods that could potentially prevent the model from learning from shortcut information only. It shows that one of the two methods is helpful in reducing learning from shortcuts.

**Audience:**

Yes

**Claims And Evidence:**

Yes

**Requested Changes:**

1. Are there ways to improve/combine existing methods to make the training more resilient to the shortcut injection?

2. What are the potential reasons that IFM does not work very well, and is there any empirical evidence to support that?

Minor:
1. Table last row third column value is unusually large and much larger than the baseline. Are there any explanations for that?

**Strengths And Weaknesses:**

Strengths:
1. The paper studies the essential problem of identifying potential caveats for contrastive learning with multiple views of the data points. The findings could be helpful for the community and intriguing for future research.

2. The paper conducts thorough experiments with solid evidence that 1) contrastive learning is prone to shortcut info and 2) some existing methods can effectively prevent that.

3. The paper gives some theoretical analysis showing that contrastive learning cannot learn the optimal task info.

Weaknesses:
1. The paper proposes a new method that can introduce shortcut info into the dataset. However, the way that the MNIST digits are injected into the image is not well-studied and tested. For example, what if the digits are injected in a way that is less detectable? In other words, there is no knob provided to control the strength of the injection.

2. The paper only evaluates existing methods for solving the shortcut problem. It does not propose new techniques or combine existing methods to improve performance. Also, the improvement for the LTD method is not that significant. It is not well understood why IFM does not give benefits.

---

> ### Author Response · Authors · 2024-06-03
> **Response to Reviewer RZvX**
>
> We thank the reviewer for the review and the encouraging suggestions. We reply to each weakness and requested change individually.
>
> *Weaknesses*:
> - The paper proposes a new method that can introduce shortcut info into the dataset. However, the way that the MNIST digits are injected into the image is not well-studied and tested. For example, what if the digits are injected in a way that is less detectable? In other words, there is no knob provided to control the strength of the injection.
>
> We thank the reviewer for pointing this out. We agree that it would be interesting to have a ‘knob’ to control the strength of the injection. However, it is unclear how we should account for this strength of injection during evaluation. In our current SVL setup, we focus on two extreme scenarios where the shortcuts are either fully present or absent. Such a setup allows us to control the shortcuts during training and evaluation and assess their impact on the learned representations. If a shortcut is less detectable during training, it becomes unclear whether the model has learned the shortcut feature or if it simply cannot detect it due to its subtlety. Future work could explore designing a new version of the SVL framework with more controllable aspects, such as varying transparency, color, size, and shape. However, that would entail defining a clear metric or method for evaluating the effects of more nuanced shortcuts.
>
> - The paper only evaluates existing methods for solving the shortcut problem. It does not propose new techniques or combine existing methods to improve performance. Also, the improvement for the LTD method is not that significant. It is not well understood why IFM does not give benefits.
>
> We thank the reviewer for bringing up this point. Indeed, our primary goal is not to introduce a new method to reduce shortcut features but rather to motivate and present the SVL framework and evaluate existing shortcut reduction methods using this framework. The evaluation helps identify areas for improvement and points out the challenges faced by current methods. Developing a new method to address the problem would be more appropriate in future work. We have emphasised the importance of this topic in our future work section.
>
>
> *Requested Changes*:
> - Are there ways to improve/combine existing methods to make the training more resilient to the shortcut injection?
>
> Thank you for this interesting suggestion. The reason we have not explored this direction in the paper is that, in this work, we focus on individual evaluation of existing shortcut reduction methods on the SVL framework. Our primary goal is to introduce the SVL framework, explore shortcut learning in contrastive VL representation learning, and assess the robustness of current shortcut reduction methods within this framework. Combining these methods would result in hybrid loss functions, making it difficult to isolate the effects of each method on shortcut reduction. Developing such combined methods would imply a new approach to solving the shortcut reduction problem, which is beyond the main objective of this paper. That said, the suggestion represents a promising direction for future work.  To address the suggestion, we have updated our future work section to highlight it.
>
> - What are the potential reasons that IFM does not work very well, and is there any empirical evidence to support that?
>
> Thank you for this question. There are several potential reasons why IFM does not work very well.
> First, IFM has primarily been evaluated in the context of self-supervised image representation learning. The dynamics of Vision-Language representation learning could differ significantly, potentially affecting IFM performance.
> Second, IFM has been tested mainly with a single encoder network and one contrastive loss. In contrast, our setup involves a dual encoder network with both image-to-text and text-to-image contrastive losses. The training dynamics in such a setup differ from those in a single encoder setup, which could explain the reduced effectiveness of IFM in our experiments.
>
> - Table last row third column value is unusually large and much larger than the baseline. Are there any explanations for that?
>
> Thank you for spotting this! This is a typo: the 4 in 421 should not have been there. We have updated the paper accordingly.

---

### Review · Reviewer_wmwY · 2024-04-30

**Summary Of Contributions:**

Summary of contributions: This paper first finds that contrastive learning in vision-language models tends to capture the "shortcut" features rather than the task-optimal features. Then, the paper proposes two methods to reduce the effect of "shortcut" features.

**Audience:**

Yes

**Broader Impact Concerns:**

no ethical concerns.

**Claims And Evidence:**

Yes

**Requested Changes:**

Please refer to the weakness

**Strengths And Weaknesses:**

Strengths: This paper did a solid theory analysis from the perspective of multi-view learning and mutual information. A synthetic shortcut experiments also report an interesting finding to demonstrate the effect of shortcuts. Extensive experiments are conducted to verify the proposed two methods.

Weakness:

1.The figure1 is not intuitive enough to illustrate the concept of "shortcuts". Authors should compare (a) and (b) and give an intuitive explanation about shortcut learning and task optimal learning.
2.There is no figure to illustrate how synthetic shortcuts are added to the image and text data. It would be better to move Figure 4 to the main body for better experiment understanding.
3. The description of the proposed two methods is too short (sections 5.1 and 5.2). The insight behind these should be stated more clearly.
4.It would be better to provide some qualitative analysis, such as giving an example with one image equipped with multiple captions, and stating what is the task optimal representations.

---

> ### Author Response · Authors · 2024-06-03
> **Response to Reviewer wmwY**
>
> We are grateful to the reviewer for the feedback and encouraging suggestions. Below, we address each weakness individually.
>
> *Weaknesses*
>
>  - The Figure 1 is not intuitive enough to illustrate the concept of "shortcuts". Authors should compare (a) and (b) and give an intuitive explanation about shortcut learning and task optimal learning.
>
> We appreciate the reviewer’s observation. We have refined the caption to provide a clearer comparison between (a) and (b) and included an intuitive explanation of the concept of shortcuts and task-optimal learning.
>
> -  There is no Figure to illustrate how synthetic shortcuts are added to the image and text data. It would be better to move Figure 4 to the main body for better experiment understanding.
>
> Thank you for this suggestion! To address it, we have added a figure (Figure 4 in Section 3), illustrating how synthetic shortcuts are added to images and captions to the main body of the paper.
>
> - The description of the proposed two methods is too short (sections 5.1 and 5.2). The insight behind these should be stated more clearly.
>
> Thank you for bringing this to our attention. In response to this feedback, we have revised sections 5.1 and 5.2 and extended the descriptions of both methods. Additionally, we have added a new section (Section 5.3), dedicated to summarising the key insights behind each method and providing a comparison between them. On top of that, to provide mathematical definitions of each method, we have added a new appendix (Appendix E).
>
> -  It would be better to provide some qualitative analysis, such as giving an example with one image equipped with multiple captions, and stating what is the task optimal representations.
>
> We thank the reviewer for highlighting this. We have added a new figure (Figure 1) that includes an example of one image with multiple captions and an explanation of task-optimal representations.

---

> > ### Comment · Reviewer_wmwY · 2024-06-05
> > **Response to authors**
> >
> > Dear authors,
> >
> > Thank you for your response. This revision presents a much more clear structure for understanding this paper.
> >
> > Actually, this paper proposes a solid framework to evaluate whether contrastive learning (CL) tends to capture "short-cut" features rather than task-optimal features.
> >
> > This paper discovers the existence of "short-cut" learning and verifies whether two current CL methods can alleviate this problem.
> >
> > I think the main contribution of this paper is proposing a meaningful and challenging problem to the machine learning community.

---

### Review · Reviewer_nQcp · 2024-05-21

**Summary Of Contributions:**

1. The authors have developed a new framework called "synthetic shortcuts for vision-language" that allows deliberate addition of shortcuts to image-caption pairs in training data. This framework helps in studying how much contrastive vision-language models (specifically CLIP and VSE++) depend on these shortcuts when they are available. By testing the models on the Flickr30k and MS-COCO benchmarks, the authors compare their performance with and without these synthetic shortcuts. The results show that both models, whether trained from scratch or fine-tuned from large pre-trained models, tend to rely heavily on these shortcut features, rather than learning the most effective representations for the tasks. This suggests that contrastive losses tend to pick up on the easy, obvious features that are shared between images and their matching captions, ignoring other important information. Therefore, the authors argue that contrastive losses alone aren't enough to learn the best representations for vision-language tasks.

2. The authors have also introduced two methods to reduce shortcut learning in their framework: latent target decoding (LTD) and implicit feature modification (IFM). Both methods improve the models' performance on evaluation tasks, but they still face challenges. The models trained with these methods don't perform as well as those trained without any synthetic shortcuts. This highlights the complexity of the framework and the need for more effective techniques to tackle shortcut learning in contrastive vision-language representation learning.

**Audience:**

Yes

**Broader Impact Concerns:**

The reliance on shortcuts in vision-language models can lead to ethical concerns in real-world applications. Models that primarily learn from shortcuts may fail to capture nuanced and critical details in images and text, potentially resulting in biased or inaccurate outcomes. For instance, in applications such as medical imaging or autonomous driving, reliance on superficial features rather than comprehensive representations could have serious, even life-threatening consequences.

Another concern is the transparency and explainability of VLMs. Models that rely on shortcuts may make decisions based on features that are not easily interpretable or explainable to users. This lack of transparency can undermine trust in AI systems, particularly in critical applications where understanding the model's decision-making process is essential.

**Claims And Evidence:**

Yes

**Requested Changes:**

1. The presentation is generally good but it could be better to provide sample cases to demonsatrate what is unique information in a sample pair, which could make the paper more readable.

2. To address the narrow focus on contrastive losses, it would be beneficial for the authors to investigate other loss functions or alternative training strategies that might mitigate shortcut learning more effectively. Incorporating a comparison of different approaches could provide a more comprehensive understanding of how to achieve task-optimal representations in vision-language models.

**Strengths And Weaknesses:**

Strengths:

1. The idea of creating a synthetic shortcuts framework is quite interesting. It allows researchers to deliberately inject shortcuts into the training data, making it possible to analyze how much vision-language models (VLMs) rely on these shortcuts. This controlled setup is a clever way to explore the weaknesses in current models and provides a clear path to understanding and improving contrastive learning.

2. The authors did a thorough job by testing their methods on well-known benchmarks like Flickr30k and MS-COCO. By evaluating two distinct models (CLIP and VSE++), both from scratch and fine-tuned, they provide a robust set of results that highlight the prevalent issues in shortcut learning. This comprehensive approach ensures that the findings are not model-specific and can be generalized across different VLMs.

3. The introduction of latent target decoding (LTD) and implicit feature modification (IFM) as methods to reduce shortcut learning is another strength. Even though these methods only partially address the problem, they still represent significant steps forward. The authors' honest assessment of the challenges these methods face underscores the complexity of the issue and the need for further research, but it also shows their potential to improve model performance.

Weakness:

1. While the methods introduced for reducing shortcut learning, such as latent target decoding (LTD) and implicit feature modification (IFM), are interesting, they only partially mitigate the problem. The paper acknowledges that these techniques do not bring the models' performance up to the level of those trained without synthetic shortcuts, indicating that more effective solutions are still needed.

2. The reliance on synthetic shortcuts to investigate the problem might limit the real-world applicability of the findings. Synthetic data does not perfectly capture the complexities and nuances of natural data. This means that the insights gained might not fully translate to scenarios where the shortcuts are not artificially introduced but are inherent in the data.

3. The paper primarily focuses on the limitations of contrastive losses in learning task-optimal representations. While this is a valid concern, it might have been beneficial to explore a broader range of loss functions or alternative training strategies. By concentrating mostly on contrastive learning, the paper may miss out on other potential avenues for improving vision-language models.

---

> ### Author Response · Authors · 2024-06-03
> **Response to Reviewer nQcp, Comment 1/2**
>
> We thank the reviewer for the review and the encouraging suggestions. We reply to each weakness and requested change individually and address a broader impact concern.
>
> *Weaknesses*:
> - While the methods introduced for reducing shortcut learning, such as latent target decoding (LTD) and implicit feature modification (IFM), are interesting, they only partially mitigate the problem. The paper acknowledges that these techniques do not bring the models' performance up to the level of those trained without synthetic shortcuts, indicating that more effective solutions are still needed.
>
> Thank you for pointing this out. Indeed, while techniques like LTD and IFM offer promising avenues for mitigating shortcut learning, our study highlights that they only partially address the issue. Our primary goal is not to introduce a new method to reduce shortcut features but rather to motivate and present the SVL framework and evaluate existing shortcut reduction methods using this framework. The evaluation helps identify areas for improvement and point out the challenges of current methods face.
>
> - The reliance on synthetic shortcuts to investigate the problem might limit the real-world applicability of the findings. Synthetic data does not perfectly capture the complexities and nuances of natural data. This means that the insights gained might not fully translate to scenarios where the shortcuts are not artificially introduced but are inherent in the data.
>
> We are grateful to the reviewer for bringing this to our attention. We agree with the reviewer that our synthetic shortcuts do not capture all the complexities and nuances of natural data. In this work, we introduce the initial version of the SVL framework, with the synthetic shortcuts being visible. However, we acknowledge the necessity of extending the SVL framework in future research to incorporate additional parameters that better match natural data. This expansion could facilitate a more nuanced understanding of shortcut learning and its implications in real-world applications. To address this concern, we have updated the future work section accordingly.
>
> - The paper primarily focuses on the limitations of contrastive losses in learning task-optimal representations. While this is a valid concern, it might have been beneficial to explore a broader range of loss functions or alternative training strategies. By concentrating mostly on contrastive learning, the paper may miss out on other potential avenues for improving vision-language models.
>
> Thank you for bringing up this point. Indeed, the focus on contrastive losses in the paper narrows the scope of the paper. This specific focus is closely tied to the second requested change below, focused on the importance of investigating a broader range of loss functions and training strategies. We address this concern in our response to the requested change. We thank the reviewer for their insight and will aim to consider a wider range of approaches in future work.

---

> > ### Author Response · Authors · 2024-06-03
> > **Response to Reviewer nQcp, Comment 2/2**
> >
> > *Requested Changes*
> >
> > - The presentation is generally good but it could be better to provide sample cases to demonstrate what is unique information in a sample pair, which could make the paper more readable.
> >
> > We appreciate the reviewer’s suggestion. We addressed it by adding a new figure (Figure 1) illustrating the concept of shared vs. caption-specific task-relevant information based on an example of an image and two associated captions. We have updated the corresponding section in the paper by providing a discussion of the concept of shared vs. unique information given the example.
> >
> > - To address the narrow focus on contrastive losses, it would be beneficial for the authors to investigate other loss functions or alternative training strategies that might mitigate shortcut learning more effectively. Incorporating a comparison of different approaches could provide a more comprehensive understanding of how to achieve task-optimal representations in vision-language models.
> >
> >  We thank the reviewer for the interesting suggestion. Initially, we intended to expand the focus of our work beyond contrastive representation learning losses. As such, we experimented with VicReg [1], a non-contrastive representation learning method; however, we were unable to reproduce the multi-modal results from the original paper for VSE++, which appears to be a known issue in the community [2].
> > To the best of our knowledge, there are currently few non-contrastive representation learning methods evaluated in a multi-modal (vision-language) setting. While there are many losses, such as masked language modelling, used in combination with contrastive losses in VL models, these losses cannot be used in isolation for representation learning with global dual-encoder image-text methods like CLIP and VSE++. Consequently, we decided to focus on contrastive losses for this work.
> >
> >
> > *Broader Impact Concerns*
> > - The reliance on shortcuts in vision-language models can lead to ethical concerns in real-world applications. Models that primarily learn from shortcuts may fail to capture nuanced and critical details in images and text, potentially resulting in biased or inaccurate outcomes. For instance, in applications such as medical imaging or autonomous driving, reliance on superficial features rather than comprehensive representations could have serious, even life-threatening consequences.
> > Another concern is the transparency and explainability of VLMs. Models that rely on shortcuts may make decisions based on features that are not easily interpretable or explainable to users. This lack of transparency can undermine trust in AI systems, particularly in critical applications where understanding the model's decision-making process is essential.
> >
> > We thank the reviewer for highlighting this. We appreciate the consideration of broader impact concerns connected to the reliance on shortcuts in VLMs raised in this statement, both in terms of ethical implications and transparency and explainability.
> >
> > To explicitly address these concerns, we added a broader impact statement to the paper, acknowledging the ethical considerations and emphasising the importance of awareness of the shortcut learning problem in the context of responsible development and deployment of VLMs.
> >
> >
> > *References*
> >
> > [1] Bardes, Adrien, Jean Ponce, and Yann LeCun. "Vicreg: Variance-invariance-covariance regularization for self-supervised learning." arXiv preprint arXiv:2105.04906 (2021).
> >
> > [2] https://github.com/facebookresearch/vicreg/issues/22

---

> ### Comment · Reviewer_nQcp · 2024-06-24
> **Response to authors**
>
> Dear authors,
>
> Thank you for your insightful rebuttal. The narrow focus on contrastive losses is recognized, and the authors intend to explore a broader range of loss functions and training strategies in future work. The added new figure (Figure 1) is very clear and easy to understand. Based on vicreg/issues/22, it is indeed unnecessary to compare your method with VSE++.
>
> The rebuttal addresses the main points.
>
> Reviewer nQcp

---

### Author Response · Authors · 2024-06-03
**Uploaded Revision of the Paper**

We would like to thank all the reviewers for their reviews and constructive feedback. We have uploaded a revised version of the paper. All changes in the text are highlighted in green. The main points addressed in the paper are:

- We improved our explanation of the concept of shared and unique caption-specific information by adding a new figure (Figure 1) and its description (Reviewer *nQcp*).
- We clarified the caption of Figure 2 (Reviewer *wmwY*).
- To illustrate how synthetic shortcuts are added to the image and text data, we included a new figure (Figure 4) (Reviewer *wmwY*).
- To clarify insights behind LTD and IFM, we extended Sections 5.1 and 5.2, added a new section (Section 5.3) dedicated to comparing them, and added a new appendix (Appendix E) describing each method mathematically (Reviewer *wmwY*).
- We added a paragraph to Section 8 discussing future work focused on extending the proposed framework and proposing new methods to mitigate the shortcut learning problem (Reviewer *RZvX)*.
- To address broader impact concerns, we included a broader impact statement (Reviewer *nQcp*).
- We fixed a typo in Table 1 (Reviewer *RZvX*).


We are happy to continue the discussion and to further address any questions or requests the reviewers might have.

Thank you,

The authors

---

### Decision · Action_Editor_5sQF · 2024-07-04

**Recommendation:** Accept as is

**Comment:**

The paper addresses the significant problem of "short-cut" learning with solid theoretical support and extensive experiments, providing valuable insights to the machine learning community. While it does not propose new or combined methods, its detailed analysis and findings are relevant and interesting. The clear presentation of evidence, including the newly added Figure 1, enhances its contribution.

**Audience:**

Some individuals in TMLR's audience would likely be interested in knowing the findings of this paper. The problem of "short-cut" learning is significant and relevant to the field, and the paper provides detailed theoretical support and extensive experiments that offer valuable insights. These findings contribute to the broader understanding of model robustness and generalization, which are critical topics in machine learning research.

**Claims And Evidence:**

Based on the reviewers' comments, the claims in the submission are supported by accurate, convincing, and clear evidence. The clear and understandable Figure 1, detailed theoretical support, and extensive experiments indicate thorough research into the problem of "short-cut" learning. The authors have also adequately addressed the main points in their rebuttal.

---

> ### Author Response · Authors · 2024-07-31
> **Response to decision by Action Editor**
>
> Dear Action Editor and Reviewers,
>
> Thank you for your time and efforts in reviewing and accepting our paper.
>
> We have uploaded the camera-ready version.
>
> Thank you once again for your contributions to this work.
>
> The authors